# Conformal Prediction in The Loop: A Feedback-Based Uncertainty Model for Trajectory Optimization

**Han Wang**
School of Automation and Intelligent Sensing
Shanghai Jiao Tong University
Shanghai 200240, China
h.wang@sjtu.edu.cn

**Chao Ning**[*]
School of Automation and Intelligent Sensing
Shanghai Jiao Tong University
Shanghai 200240, China
chao.ning@sjtu.edu.cn

## Abstract

Conformal Prediction (CP) is a powerful statistical machine learning tool to construct uncertainty sets with coverage guarantees, which has fueled its extensive adoption in generating prediction regions for decision-making tasks, e.g., Trajectory Optimization (TO) in uncertain environments. However, existing methods predominantly employ a sequential scheme, where decisions rely unidirectionally on the prediction regions, and consequently the information from decision-making fails to be fed back to instruct CP. In this paper, we propose a novel Feedback-Based CP (Fb-CP) framework for shrinking-horizon TO with a joint risk constraint over the entire mission time. Specifically, a CP-based posterior risk calculation method is developed by fully leveraging the realized trajectories to adjust the posterior allowable risk, which is then allocated to future times to update prediction regions. In this way, the information in the realized trajectories is continuously fed back to the CP, enabling attractive feedback-based adjustments of the prediction regions and a provable online improvement in trajectory performance. Furthermore, we theoretically prove that such adjustments consistently maintain the coverage guarantees of the prediction regions, thereby ensuring provable safety. Additionally, we develop a decision-focused iterative risk allocation algorithm with theoretical convergence analysis for allocating the posterior allowable risk which closely aligns with Fb-CP. Furthermore, we extend the proposed method to handle distribution shift. The effectiveness and superiority of the proposed method are demonstrated through benchmark experiments.

## 1 Introduction

In recent years, Trajectory Optimization (TO) has recently garnered significant attention in the academic community Pan et al. [2024] and has achieved significant success in fields such as autonomous driving Zhou et al. [2020], autonomous surface vessels Tsolakis et al. [2024], and coverage control Davis et al. [2016]. However, collision-free TO in uncertain environments is a formidable challenge, because the intentions of obstacles are unknown. A crucial aspect of collision avoidance involves predicting obstacle trajectories. Existing trajectory prediction tools are unable to predict fully accurate trajectories. Therefore, a common approach is to generate the $(1 - \alpha)$-coverage prediction regions of the obstacle trajectories. If these regions contain the true trajectories with a probability of at least $1 - \alpha$, they are considered *valid*. The key to probabilistic collision-free TO lies in adjusting the prediction regions while remaining valid to improve the trajectory performance.

Conformal Prediction (CP) is an attractive framework to produce prediction regions with finite-sample guarantees of validity Vovk et al. [2005], Shafer and Vovk [2008]. Without imposing any assumptions

---

[*]Corresponding author.

39th Conference on Neural Information Processing Systems (NeurIPS 2025).

about prediction algorithms and data distributions, CP utilizes a calibration dataset to obtain a valid prediction region for test data. Owing to its simplicity and versatility, CP and its variants have been widely applied in various safety-critical applications, such as probabilistic collision-free TO Lindemann et al. [2023], reliable estimation of graph neural networks H. Zargarbashi et al. [2023] and language modeling Quach et al. [2024].

However, there is a disconnect between existing research on CP theory and CP application for decision-making. On the side of CP theory, most existing work primarily focuses on upstream data, developing new CP algorithms to enhance prediction performance, such as addressing distributional shifts Gibbs and Candes [2021], performing multi-step time forecasting Sun and Yu [2023], and improving the efficiency of prediction regions Bai et al. [2022]. There is a lack of CP algorithms focused on enhancing the performance of downstream decisions. On the side of CP application for decision-making, most existing work embeds the CP into decision-making pipelines as a method for generating prediction regions, and employs a sequential approach, i.e. the prediction region is first computed using CP and then the decision depends unidirectionally on the prediction region without considering the favorable impact of the decision on the prediction region. However, this information channel blockage from the decision-making to the CP seriously prevents the CP from leveraging the information of past decisions to boost the performance of future decisions. Therefore, there is a pressing research need to develop a closed-loop framework that seamlessly integrates CP with decision-making, fully exploiting the information of past decisions to adjust prediction regions and thereby remarkably enhance the performance of future decisions.

To fill the aforementioned research gap, we propose a Feedback-Based CP (Fb-CP) framework for shrinking-horizon TO in uncertain environments and the collision risk over the total mission time is constrained at all times. The proposed framework leverages CP to construct the prediction regions of obstacle positions and adjusts these regions online in a closed-loop fashion while ensuring coverage guarantees, i.e. validity. In particular, we propose a novel posterior probability calculation method to obtain the posterior probability of collision conditional on realized trajectories. The posterior collision probability is then used to adjust the allowable collision risk, which is allocated to future times to yield prediction regions. In this manner, information from past trajectories is fed back to the CP through the posterior probability calculation, guiding the feedback-based adjustments of the prediction regions. Such adjustments in Fb-CP not only offer provable performance improvements but also consistently maintain the validity of the prediction regions. With the adjusted prediction regions, the trajectory is obtained by solving the resulting TO problem. Additionally, we further propose a decision-focused risk allocation method, i.e. Iterative Risk Allocation (IRA), which aims to optimize the trajectory performance by iteratively allocating the allowable risk to future times while enjoying the convergence guarantee. We highlight the main contributions of our work below.

- We propose, for the first time in the literature Fb-CP, a general uncertainty quantification framework closely associated with downstream decision-making which enables the adjustment of prediction regions using the feedback information embedded in decisions.

- We prove that 1) the feedback-based adjustments in Fb-CP do not compromise the coverage guarantees of prediction regions, and 2) Fb-CP offers guarantees for decision-making performance improvement. In other words, Fb-CP enjoys both validity and superior performance.

- We propose a decision-focused risk allocation algorithm with theoretical convergence analysis for Fb-CP, which optimizes the risk allocation to enhance decision-making performance.

- We extend Fb-CP to handle distribution shift by applying a weighting scheme to the test and calibration data and demonstrate its effectiveness.

## 2 Related work

**Conformal Prediction.** Conformal prediction originated in the early work Vovk et al. [1999, 2005], Shafer and Vovk [2008] to generate the prediction region. The salient advantage of CP lies in its ability to offer coverage guarantees regardless of prediction algorithms and data distributions. Most recently, various variants of CP have been developed to handle upstream data with different characteristics H. Zargarbashi et al. [2024], Liu et al. [2024] or to produce prediction regions in a wide array of forms Angelopoulos et al. [2024], Auer et al. [2023]. In response to the distribution shift in the upstream data, ACI Gibbs and Candes [2021], Podkopaev et al. [2024], Zaffran et al. [2022] and EnbPI Xu and Xie [2021, 2023] developed CP through online learning and sliding window,

respectively, and achieved asymptotic validity. In the context of multi-step time series forecasting, Sun and Yu [2023] combined CP with copula to propose the CopulaCPTS, while Cleaveland et al. [2024] employed an optimization-based method. Zhou et al. [2024b] presented a new conformal method for time series forecasting. In addition, numerous studies focused on improving the efficiency of the prediction region by changing the region shape Xu et al. [2024], minimizing the region length Kiyani et al. [2024], or optimizing the region construction function Bai et al. [2022]. Note that the prediction regions are typically utilized by downstream tasks in a sequential manner. However, the above research work primarily aims to enhance the predictive performance of CP rather than directly improving the performance of downstream decision-making.

**Decision-making with CP.** Many studies have focused on the integration of CP with decision-making. Vovk and Bendtsen [2018] was the first to the method of CP to make it applicable to decision-making. Additionally, Fannjiang et al. [2022] proposed a CP with coverage guarantee under one-step Feedback Covariate Shift (FCS), in which the test data depend on the training data. Building upon the aforementioned work, Prinster et al. [2024] refined its theoretical framework and extended it to multistep FCS. Furthermore, Lekeufack et al. [2024] introduced a conformal decision theory, which follows the ACI concept to directly provide provable statistical guarantees of having low risk for decisions made based on uncertainty-aware predictions.

**TO using CP.** Lindemann et al. [2023] and Strawn et al. [2023] applied CP to the safe planning for single-robot systems, while Muthali et al. [2023] and Kuipers et al. [2024] extended it to multi-robot systems. Additionally, Dixit et al. [2023] and Zhou et al. [2024a] employed the ACI to address the obstacle trajectory distribution shift. Stamouli et al. [2024] proposed a novel nonconformity score for shrinking-horizon TO. All the above methods directly employ CP in a sequential way to generate prediction regions. Nevertheless, the performance of realized trajectories has yet to be conveyed to the upstream CP as feedback information to adjust the prediction regions, which has the great potential to further boost the performance of trajectory.

## 3 Problem formulation and background

### 3.1 Problem formulation

Consider a discrete-time nonlinear dynamics as follows.

$$x_{t+1} = f(x_t, u_t), \quad x_0 = x_{init} \tag{1}$$

where $x_t \in \mathcal{X} \subseteq \mathbb{R}^{n_x}$ and $u_t \in \mathcal{U} \subseteq \mathbb{R}^{n_u}$ are the state and control at time $t = 0, ..., T$, respectively, and $T \geq 1$ is the total mission time. The sets $\mathcal{U}$ and $\mathcal{X}$ represent the admissible sets of control inputs and system states, respectively. The function $f : \mathbb{R}^{n_x} \times \mathbb{R}^{n_u} \to \mathbb{R}^{n_x}$ represents the system dynamics and $x_{init}$ is the initial state. For brevity, let $x_{t_1:t_2} := (x_{t_1}, ..., x_{t_2})$ and $u_{t_1:t_2} := (u_{t_1}, ..., u_{t_2})$ denote the state and control sequences from $t_1$ to $t_2$, respectively. The system operates in an environment with $M$ dynamic obstacles with a priori unknown trajectories. Let $Y_t := (Y_{t,1}, ..., Y_{t,M})$ represent the joint obstacle state at time $t$, where $Y_{t,j} \in \mathbb{R}^{n_p}$ denotes the state of obstacle $j$ at time $t$. The dynamic of the above joint obstacle system can be expressed as follows.

$$Y_{t+1} = g(Y_{t-h}, ..., Y_{t-1}) + \omega_t \tag{2}$$

where $h$ is the window length, $g(\cdot)$ represents the model learned from the historical data of the obstacle using machine learning tool, i.e., Long Short-Term Memory (LSTM). Naturally, readers may choose to adopt more powerful predictors for improved accuracy, such as Salzmann et al. [2020], Yuan et al. [2021], Salzmann et al. [2023], Yuan and Kitani [2020]. $\omega_t$ captures the modeling error. Additionally, we define $Y := (Y_0, ..., Y_T)$ as the entire trajectories of the obstacles, which is generated by sampling the initial state $Y_0$ from an unknown probability distribution $\mathcal{D}$ and by evolving it based on the dynamics (2). The system can observe the joint obstacle states $Y_0, ..., Y_t$, when making the decision at time $t$. We assume the availability of an offline dataset as follows.

**Assumption 3.1** *We have a calibration dataset $D_{cal} := \{Y^{(1)}, ..., Y^{(N)}\}$, where each of the $N$ joint obstacle trajectories is generated by independently sampling an initial state $Y_0^{(i)}$ from $\mathcal{D}$ and evolving it based on its ground truth dynamics.*

With Assumptions 3.1, we can conclude that the real joint obstacle trajectory $Y$ and the $N$ available joint obstacle trajectories $Y^{(i)}$ are independently and identically distributed (i.i.d.), and are also

exchangeable. Assumption 3.1 is not restrictive in practice, e.g. the historical trajectories of obstacles. In Appendix G, we introduce an extension to address the case where a shift exists between the initial state distributions of calibration data and test data, and the error $\omega_t$ represents state-dependent noise rather than being i.i.d..

We focus on the TO problem whose objective is to find the sequences $x_{1:T}$ and $u_{0:T-1}$ that minimize the cost function $J(x_{1:T}, u_{0:T-1})$ subject to the dynamics and constraints. The TO is performed in a shrinking-horizon fashion, with the optimization problem at time $t$ formulated as follows.

$$\min_{x_{t+1:T}, u_{t:T-1}} \quad J(x_{t+1:T}, u_{t:T-1}) = l_T(x_T) + \sum_{\tau=t}^{T-1} l_\tau(x_\tau, u_\tau) \tag{3a}$$

$$s.t. \quad x_{\tau+1} = f(x_\tau, u_\tau), \qquad \forall \tau = t, ..., T-1 \tag{3b}$$

$$x_\tau \in \mathcal{X}, \qquad \forall \tau = t+1, ..., T \tag{3c}$$

$$u_\tau \in \mathcal{U}, \qquad \forall \tau = t, ...T-1 \tag{3d}$$

$$\mathbb{P}\left\{ \bigcap_{\tau=1}^{T} \{c(x_\tau, Y_\tau) \geq 0\} \right\} \geq 1 - \alpha \tag{3e}$$

where $\mathbb{P}\{X\}$ denotes the probability of event $X$, the constraint function $c := \mathbb{R}^{n_x} \times \mathbb{R}^{n_p} \to \mathbb{R}$ is $L$-Lipschitz continuous, which can encode various tasks, such as collision avoidance. Due to the uncertainty of the joint obstacle state $Y_\tau$, we impose the joint chance constraint (3e) with failure probability $\alpha \in (0, 1)$ to ensure that the joint probability of satisfying the constraint over the total mission time is no less than $1 - \alpha$. To ensure the initial feasibility of the TO problem, we assume that the initial state satisfies the constraint, i.e. $c(x_0, Y_0) \geq 0$, with probability 1.

### 3.2 Conformal prediction

In this subsection, we provide a brief introduction to the theoretical results for CP and refer readers to Angelopoulos and Bates [2021] for a thorough introduction. Given $N + 1$ exchangeable random variables $R^{(0)}, R^{(1)}, ..., R^{(N)}$ which is usually referred to as the nonconformity score, CP aims to find a probabilistic upper bound for $R$ based on $R^{(1)}, ..., R^{(N)}$ such that $R$ is less than this upper bound with high probability. In practice, $R^{(0)}$ represents the test data point, while $R^{(1)}, ..., R^{(N)}$ denote the calibration dataset. Formally, the central idea behind CP is summarized in Lemma A.1 provided in Appendix A.1.

## 4 Feedback-based conformal prediction

The challenge in solving the TO problem (3) lies in the computation of the joint probability (3e). Existing literature predominantly employs a sequential way of using CP, i.e. the prediction regions of obstacle positions are first computed based on the failure probability $\alpha$, and then the decision of TO depends one-way on the prediction regions. However, it is important to note that in the shrinking-horizon TO framework, at time $t$ the past decisions $x_{0:t}$ are available and typically contain rich information that can instrumentally assist in refining the prediction regions at subsequent time steps, thereby considerably improving the performance of TO. Therefore, we propose a novel Feedback-based Conformal Prediction (Fb-CP). Fb-CP not only exploits the feedback information provided by realized trajectories to perform closed-loop adjustments of the prediction regions but also maintains coverage guarantees. To begin with, the joint chance constraint (3e) can be reformulated as a set of individual chance constraints and a total risk constraint by using Boole's inequality as follows.

$$\mathbb{P}\left\{ \bigcap_{\tau=1}^{T} \{c(x_\tau, Y_\tau) \geq 0\} \right\} \geq 1 - \alpha \impliedby \left\{ \begin{array}{l} \mathbb{P}\{c(x_\tau, Y_\tau) \geq 0\} \geq 1 - \alpha_\tau, \ \forall \tau = 1, ..., T \\ \sum_{\tau=1}^{T} \alpha_\tau \leq \alpha \end{array} \right. \tag{4}$$

The risk $\alpha_\tau$ at each time can be initially allocated uniformly at time $t = 0$, i.e. $\alpha_\tau = \alpha/T$, and remains constant throughout the shrinking-horizon TO process, as in Lindemann et al. [2023]. However, at time $t$, the system states $x_\tau$ for $\tau \leq t$ are available, which grants us to compute the posterior probability $\beta_\tau = \mathbb{P}\{c(x_\tau, Y_\tau) > 0 | x_\tau\}$ and the permissible risk for future times, which is then used to adjust the prediction regions. Using the information in the realized trajectories, the feedback-based adaptation of the prediction regions tremendously reduces the conservatism of trajectory online while ensuring coverage guarantees. In Subsection 4.1, we present the individual chance constraint reformulation using the prediction regions derived. In Subsection 4.2, we present a CP-based method for calculating $\beta_\tau$. We reformulate the TO problem in Subsection 4.3.

## 4.1 Constraint reformulation using conformal prediction region

We randomly divide the calibration dataset $D_{cal}$ into two subsets $D_{cal}^1$ and $D_{cal}^2$ with $K$ and $L$ joint obstacle trajectories, respectively, where $K + L = N$. Without loss of generality, we reassign indices to the joint obstacle trajectories as $D_{cal}^1 := \{Y^{(1)}, ..., Y^{(K)}\}$ and $D_{cal}^2 := \{Y^{(K+1)}, ..., Y^{(K+L)}\}$. At time $t$, we can obtain the prediction of the joint obstacle state $\hat{Y}_{\tau|t}$ for all future time $\tau = t+1, ..., T$ using $g(\cdot)$ described in (2). Similarly, the prediction $\hat{Y}_{\tau|t}^{(i)}$ for each trajectory $Y^{(i)}$ in $D_{cal}^1$ is derived by using the same method. We define the nonconformity score as follows.

$$R_{\tau|t} = \|Y_\tau - \hat{Y}_{\tau|t}\|, \quad R_{\tau|t}^{(i)} = \|Y_\tau^{(i)} - \hat{Y}_{\tau|t}^{(i)}\| \quad \forall i = 1, ..., K \tag{5}$$

Note that $Y_\tau, Y_\tau^{(1)}, ..., Y_\tau^{(K)}$ are exchangeable and the prediction function $g(\cdot)$ is trained from $D_{train}$ independent of $D_{cal}^1$. Therefore, given an allocated risk $\alpha_\tau$ for future time $\tau$, the random variables $R_{\tau|t}, R_{\tau|t}^{(1)}, ..., R_{\tau|t}^{(K)}$ are exchangeable and the prediction region with coverage guarantee is derived according to Lemma A.1 as follows.

$$\mathbb{P}\{\|Y_\tau - \hat{Y}_{\tau|t}\| \le C_{\tau|t}^{1-\alpha_\tau}\} \ge 1 - \alpha_\tau \tag{6a}$$

$$C_{\tau|t}^{1-\alpha_\tau} = Quantile_{1-\alpha_\tau}(R_{\tau|t}^{(1)}, ..., R_{\tau|t}^{(K)}, \infty) \tag{6b}$$

Based on the $(1 - \alpha_\tau)$-coverage prediction region $\{y : \|y - \hat{Y}_{\tau|t}\| \le C_{\tau|t}^{1-\alpha_\tau}\}$, the individual chance constraint in (4) can be reformulated as the following lemma proven in Appendix B.1.

**Lemma 4.1** *(chance constraint) If Assumption 3.1 holds, the constraint function $c$ is L-Lipschitz continuous and $c(x_\tau, \hat{Y}_{\tau|t}) \ge LC_{\tau|t}^{1-\alpha_\tau}$ is satisfied where $C_{\tau|t}^{1-\alpha_\tau}$ is calculated by (6b), then the individual chance constraint $\mathbb{P}\{c(x_\tau, Y_\tau) \ge 0\} \ge 1 - \alpha_\tau$ is satisfied.*

For a general collision avoidance chance constraint of the form $\mathbb{P}\{\|x_\tau - Y_\tau\| - r \ge 0\} \ge 1 - \alpha_\tau$, Lemma 4.1 effectively converts it into a deterministic constraint that requires the distance between $x_\tau$ and the predicted location $\hat{Y}_{\tau|t}$ to exceed the inflated radius derived from the prediction error, i.e., $\|x_\tau - Y_\tau\| - r - C_{\tau|t}^{1-\alpha_\tau} \ge 0$.

## 4.2 Posterior probability conditional on past decisions

At time $t$, the states $x_\tau^*$ for all past time $\tau = 1, ..., t$ are deterministic and available to the trajectory optimizer. We assume that $x_\tau^*$ is the true system state at time $\tau$. Note that $x_\tau^*$ is a feasible solution to the TO problem (3) at time $\tau - 1$ with the reformulated constraints through Lemma 4.1. Therefore, the individual chance constraint $\mathbb{P}\{c(x_\tau^*, Y_\tau) \ge 0\} \ge 1 - \alpha_\tau$ is satisfied at time $\tau - 1$ and will be naturally satisfied for all time $\tau' \ge \tau - 1$. However, the constraint violation probability $\alpha_\tau$ is a priori probability allocated before time $\tau$ that tends to overestimate the violation probability and thus leads to conservative results. Fortunately, the determined $x_\tau^*$ allows us to compute the posterior probability of constraint violation $\beta_\tau$, which, as we theoretically prove, is less than $\alpha_\tau$ with high probability. The risk redundancy between $\alpha_\tau$ and $\beta_\tau$ can be allocated across future times. In this way, the information embedded in $x_\tau^*$ is fed back from the decision-making to the CP to readjust the prediction region and to achieve a trajectory with notably improved performance. To compute $\beta_\tau$ using Lemma A.1, we propose a novel nonconformity score as follows.

$$S_\tau = c(x_\tau^*, \hat{Y}_{\tau|\tau-1} + \omega_\tau) = c(x_\tau^*, Y_\tau), \quad S_\tau^{(i)} = c(x_\tau^*, \hat{Y}_{\tau|\tau-1} + \omega_\tau^{(i)}) \quad \forall i = 1, ..., K + L \tag{7}$$

where $\omega_\tau^{(i)}$ is the modeling error of the joint obstacle trajectory $Y^{(i)}$ at time $\tau$, which can be obtained through $\omega_\tau^{(i)} = Y_\tau^{(i)} - g(Y_{\tau-1}^{(i)})$. We note that $\omega_\tau, \omega_\tau^{(1)}, ..., \omega_\tau^{(K+L)}$ are i.i.d., and if $x_\tau^*$ is fixed and independent of $\omega_\tau, \omega_\tau^{(1)}, ..., \omega_\tau^{(K+L)}$, the random variables $S_\tau, S_\tau^{(1)}, ..., S_\tau^{(K+L)}$ are also exchangeable. However, as $x_\tau^*$ is derived through the TO problem (3) at time $\tau - 1$, it depends on $D_{cal}^1$ and the random variables $S_\tau, S_\tau^{(1)}, ..., S_\tau^{(K)}$ are no longer exchangeable. Therefore, we only use the subset $D_{cal}^2$, i.e. $S_\tau^{(K+1)}, ..., S_\tau^{(K+L)}$, to compute $\beta_\tau$. The upper bound of the posterior violation probability $\beta_\tau$ is computed via the following lemma, whose proof is given in Appendix B.2.

**Lemma 4.2** *(posterior probability calculation) Assume that $x_\tau^*$ is the true state of the system at time $\tau$, then the upper bound of the posterior violation probability at time $\tau$ is as follows.*

$$\mathbb{P}\{c(x_\tau^*, Y_\tau) < 0\} \leq \beta_\tau = \left(1 + \sum_{i=1}^{L} \mathbb{I}\left(S_\tau^{(K+i)} < 0\right)\right)/(1+L) \tag{8}$$

*where $\mathbb{I}(\cdot)$ is the indicator function.*

Lemma 4.2 essentially computes the posterior probability by evaluating the fraction of calibration set $D_{cal}^2$ samples whose actual trajectories would collide with the given realized position $x_\tau^*$. Some might raise the concern that $\beta_\tau$ could be higher than $\alpha_\tau$, which could result in a more conservative trajectory when using $\beta_\tau$ in subsequent times. However, we can restricts the upper bound of the expectation of $\beta_\tau$ in Corollary A.2 provided in Appendix A.2. Additionally, we report the empirical observation that our method consistently tends to improve performance in practice, which is further discussed in Remark A.3 provided in Appendix A.2.

### 4.3  Optimization problem reformulation

By making use of the joint chance constraint reformulation (4), the individual constraint reformulation in Lemma 4.1 and the posterior probability in Lemma 4.2, the TO (3) is transformed as follows.

$$\min_{x_{t+1:T}, u_{t:T-1}, \alpha_{t+1:T}} J(x_{t+1:T}, u_{t:T-1}) = l_T(x_T) + \sum_{\tau=t}^{T-1} l_\tau(x_\tau, u_\tau) \tag{9a}$$

$$s.t. \quad (3b) - (3d) \tag{9b}$$

$$c(x_\tau, \hat{Y}_{\tau|t}) \geq LC_{\tau|t}^{1-\alpha_\tau}, \quad \forall \tau = t+1, ..., T \tag{9c}$$

$$\alpha_\tau \geq 0, \quad \forall \tau = t+1, ..., T \tag{9d}$$

$$\sum_{\tau=t+1}^{T} \alpha_\tau \leq \alpha - \sum_{\tau=0}^{t} \beta_\tau \tag{9e}$$

where Constraint (9c) ensures the satisfaction of individual chance constraints (4) for future times $\tau = t+1, ..., T$ through Lemma 4.1, and $C_\tau^{1-\alpha}$ is calculated by (6b). Constraint (9d) is imposed to ensure the non-negativity of $\alpha_\tau$. Constraint (9e) is the most important part for feedback-based adjustments of the prediction region and online performance enhancement of the optimized trajectory. It is derived by replacing $\alpha_\tau$ for past time $\tau = 1, ..., t$ in the total risk constraint (4) with $\beta_\tau$ calculated through Lemma 4.2. The information embedded in past decisions $x_1^*, ..., x_t^*$ influences the future values of $\alpha_{t+1}, ..., \alpha_T$ through the calculation of $\beta_1, ..., \beta_t$ thereby reshaping the prediction region of CP in an end-to-end way. Based on Corollary A.2 and Remark A.3, $\beta_\tau$ is highly likely to be less than $\alpha_\tau$ in practice. Consequently, using $\beta_\tau$ grants more risk to be reserved for future times, resulting in much compact prediction regions and tremendously improved optimization performance.

However, it is important to note that $C_{\tau|t}^{1-\alpha_\tau}$ depends on $\alpha_\tau$ and $D_{cal}^1$. Consequently, treating $\alpha_{t+1:T}$ as decision variables alongside $x_{t+1:T}$ and $u_{t:T-1}$ would make the optimization problem (9) computationally demanding to solve for larger values of $T$ and $K$. Therefore, we will present an allocation method for $\alpha_{t+1:T}$ that aligns with the optimization problem (9) in the next section.

## 5  Shrinking-horizon trajectory optimization using Fb-CP

The shrinking-horizon TO framework using Fb-CP is illustrated in Figure 1. The information in $x_{0:t}^*$ guides the feedback-based adjustments of the size of the prediction regions $C_{\tau|t}^{1-\alpha_\tau}$ through posterior probability calculations. Solving the TO problem (9) is divided into two steps: 1) risk allocation and 2) TO with the fixed $\alpha_{t+1:T}$. The TO problem (9) with the fixed $\alpha_{t+1:T}$ is formalized as follows.

$$\min_{x_{t+1:T}, u_{t:T-1}} J(x_{t+1:T}, u_{t:T-1}) \quad s.t. (9b) - (9c) \tag{10}$$

The problem (10) can be readily solved to obtain $x_{t+1:T}^*$ and $u_{t:T-1}^*$ and only the first system input $u_t^*$ is implemented as the control input. Therefore, as the actual time $t$ progresses, the optimization horizon gradually shrinks. For the risk allocation, a general approach is the Average-based Risk Allocation (ARA), i.e. the allocable risk is evenly allocated across future times at time $t$ below.

$$\alpha_\tau = (\alpha - \sum_{\tau=0}^{t} \beta_\tau)/(T-t) \; \forall \tau = t+1, ..., T \tag{11}$$

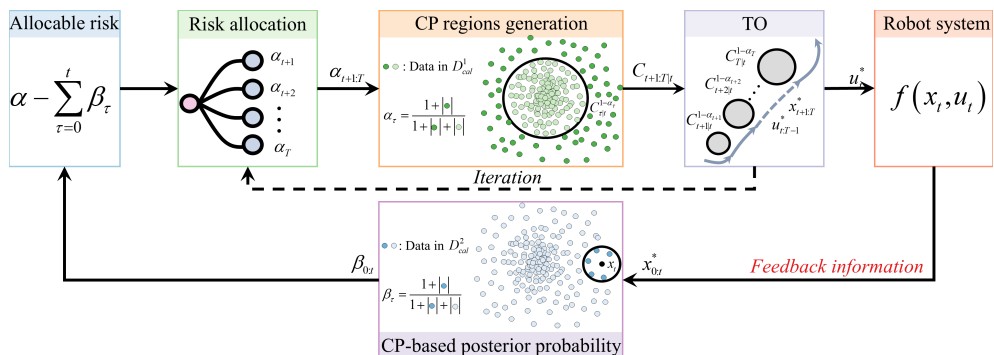

Figure 1: Shrinking-horizon trajectory optimization framework using Fb-CP.

Although the ARA method has the advantage of computational efficiency, the fixed proportion allocation significantly diminishes the flexibility in modifying the prediction regions for future times. Therefore, we extend the IRA Ono and Williams [2008] to the Fb-CP. We refer to the TO problem with the fixed $\alpha_{t+1:T}$ (10) and the risk allocation problem as the lower-stage and the upper-stage problem, respectively. The system states $x^*_{t+1:T}$ and inputs $u^*_{t:T-1}$, as well as the risk allocation $\alpha_{t+1:T}$ are obtained by iteratively solving the lower and upper-stage problems. We denote the feasible region of (10) with fixed $\alpha_{t+1:T}$ as $\mathcal{R}_t(\alpha_{t+1:T})$. The upper-stage problem is formally stated below.

$$\min_{\alpha_{t+1:T}} \quad J^*(\alpha_{t+1:T}) \tag{12a}$$

$$s.t. \alpha_\tau \geq 0, \qquad \forall \tau = t+1, ..., T \tag{12b}$$

$$\sum_{\tau=t+1}^{T} \alpha_\tau \leq \alpha - \sum_{\tau=0}^{t} \beta_\tau \tag{12c}$$

$$\alpha_{t+1:T} \in \{\alpha_{t+1:T} : \exists (x, u) \in \mathcal{R}_t(\alpha_{t+1:T})\} \tag{12d}$$

where $J^*(\alpha_{t+1:T})$ is the optimal objective function of (10) given $\alpha_{t+1:T}$. If a risk allocation $\alpha_{t+1:T}$ satisfies (12b)-(12d), then we refer to $\alpha_{t+1:T}$ a feasible risk allocation. However, the lower-stage problem (12) is challenging to solve due to the computational complexity arising from its objective (12a) and Constraint (12c). To solve (12) efficiently, we introduce a descent algorithm, i.e. IRA for Fb-CP which is based on the monotonicity of $J^*(\alpha_{t+1:T})$ below, which is proven in Appendix B.4.

**Lemma 5.1** *(monotonicity of $J^*$) At time t, the following inequalities always hold.*

$$\frac{\partial J^*(\alpha_{t+1:T})}{\partial \alpha_\tau} \leq 0 \quad \forall \tau = t+1, ..., T \tag{13}$$

The monotonicity of $J^*$ in Lemma 5.1 shows that increasing the allocated risk $\alpha_\tau$ at any time step will strictly decrease the optimal cost of the trajectory optimizer. This insight enables our risk reallocation strategy: by reducing redundant risk at inactive time steps and reallocating it to active ones, we can lower the overall optimal cost.

We assume that $\alpha^n_{t+1:T}$ represents the feasible risk allocation obtained after the $n$th iteration at time $t$. IRA aims to obtain a feasible risk allocation $\alpha^{n+1}_{t+1:T}$ in the $(n+1)$th iteration such that $J^*(\alpha^{n+1}_{t+1:T}) \leq J^*(\alpha^n_{t+1:T})$. In the $(n+1)$th iteration, IRA first solves the lower-stage problem (10) using $\alpha^n_{t+1:T}$ to obtain the optimal solution $x^n_{t+1:T}$ and $u^n_{t:T-1}$. Subsequently, based on $x^n_{t+1:T}$, Constraint (9c) in the lower-stage problem (10) is categorized into active and inactive constraints. The active and inactive constraint sets are formally defined as $\mathcal{I}_{act} := \{\tau : c(x^n_\tau, \hat{Y}_{\tau|t}) = LC^{1-\alpha^n_\tau}_{\tau|t}, \tau = t+1, ..., T\}$ and $\mathcal{I}_{ina} := \{\tau : \tau \notin \mathcal{I}_{act}, \tau = t+1, ..., T\}$, respectively. In summary, IRA consists of two steps: 1) tightening the inactive constraints and 2) relaxing the active constraints.

Tightening the inactive constraints is first implemented to construct $\widetilde{\alpha}^n_{t+1:T}$ from $\alpha^n_{t+1:T}$. Specifically, for $\tau \in \mathcal{I}_{act}$, set $\widetilde{\alpha}^n_\tau = \alpha^n_\tau$. Based on the definition of $C^{1-\alpha_\tau}_{\tau|t}$ (6b), $C^{1-\alpha_\tau}_{\tau|t}$ is non-increasing with respect to $\alpha_\tau$ for the fixed $D^1_{cal}$. Thus for $\tau \in \mathcal{I}_{ina}$, we choose $\widetilde{\alpha}^n_\tau \leq \alpha^n_\tau$ so that

$$c(x^n_\tau, \hat{Y}_{\tau|t}) \geq LC^{1-\widetilde{\alpha}^n_\tau}_{\tau|t} \geq LC^{1-\alpha^n_\tau}_{\tau|t} \tag{14}$$

Based on (14), it can be deduced that $(x_{t+1:T}^n, u_{t:T-1}^n) \in \mathcal{R}_t(\widetilde{\alpha}_{t+1:T}^n) \subseteq \mathcal{R}_t(\alpha_{t+1:T}^n)$. Therefore, the optimal solution $(x_{t+1:T}^n, u_{t:T-1}^n)$ for $\alpha_{t+1:T}^n$ is also the optimal solution for $\widetilde{\alpha}_{t+1:T}^n$, and thus $J^*(\alpha_{t+1:T}^n) = J^*(\widetilde{\alpha}_{t+1:T}^n)$. Finally, it is straightforward to show that $\widetilde{\alpha}_{t+1:T}^n$ is a feasible risk allocation, because (i) (12b) follows from (14) and the fact that when $\alpha_\tau \to 0$, $C_{\tau|t}^{1-\alpha_\tau} \to \infty$; (ii) (12c) is satisfied since $\sum_{\tau=t+1}^T \widetilde{\alpha}_\tau^n \leq \sum_{\tau=t+1}^T \alpha_\tau^n \leq \alpha - \sum_{\tau=1}^t \beta_\tau$; (iii) (12d) is satisfied because $(x_{t+1:T}^n, u_{t:T-1}^n)$ is feasible for $\widetilde{\alpha}_{t+1:T}$. The specific construction of $\widetilde{\alpha}_\tau^n$ is as follows.

$$\widetilde{\alpha}_\tau^n = \begin{cases} \alpha_\tau^n, & \tau \in \mathcal{I}_{act} \\ (1-\eta)\alpha_\tau^n + \eta\underline{\alpha}_\tau^n, & \tau \in \mathcal{I}_{ina} \end{cases} \tag{15}$$

where $\eta \in (0,1)$ is the step size and $\underline{\alpha}_\tau^n$ is the lower bound of $\widetilde{\alpha}_\tau^n$ calculated as in Lemma 5.2.

**Lemma 5.2** *(constraint tightening) Assume that $x_{t+1:T}^n$ is feasible for the problem (10) with $\alpha_{t+1:T}^n$ and $\alpha_{t+1:T}^n < 1$. For $\tau \in \mathcal{I}_{ina}$, the lower bound of $\widetilde{\alpha}_\tau^n$ while satisfying (14) is as follows.*

$$\underline{\alpha}_\tau^n = \left(1 + \sum_{i=1}^K \mathbb{I}\left(c(x_\tau^n, \hat{Y}_{\tau|t}) < LR_{\tau|t}^{(i)}\right)\right) / (1+K) \tag{16}$$

*Furthermore, it is deterministic that $\underline{\alpha}_\tau^n \leq \alpha_\tau^n$.*

Lemma 5.2 is proven in Appendix B.5. Then $\alpha_{t+1:T}^{n+1}$ is constructed from $\widetilde{\alpha}_{t+1:T}^n$ to relax the active constraints as follows.

$$\alpha_\tau^{n+1} = \begin{cases} \widetilde{\alpha}_\tau^n + (\alpha - \sum_{\tau=1}^t \beta_\tau - \sum_{\tau=t+1}^T \widetilde{\alpha}_\tau^n)/N_{act}, & \tau \in \mathcal{I}_{act} \\ \widetilde{\alpha}_\tau^n, & \tau \in \mathcal{I}_{ina} \end{cases} \tag{17}$$

where $N_{act}$ represents the number of elements in the set $\mathcal{I}_{act}$. It can be easily verified that $\alpha_{t+1:T}^{n+1}$ satisfies (12b)-(12d), and thus $\alpha_{t+1:T}^{n+1}$ is a feasible risk allocation. Note that $\alpha_\tau^{n+1} \geq \widetilde{\alpha}_\tau^n$ since $\widetilde{\alpha}_\tau^n$ satisfies (12c). Therefore, the following inequality is obtained by implying Lemma 5.1.

$$J^*(\alpha_{t+1:T}^{n+1}) \leq J^*(\widetilde{\alpha}_{t+1:T}^n) = J^*(\alpha_{t+1:T}^n) \tag{18}$$

By recursively constructing $\alpha_{t+1:T}^1, ..., \alpha_{t+1:T}^n$ in this manner, $J^*$ monotonically decreases. The algorithm of Fb-CP using IRA at time $t$ isis delineated in Algorithm 5. Note that at time $t = 0$, the input parameter $\alpha_{0:T}$ is initialized and $\epsilon$ is a given small tolerance. At time $t$, the robot first obtains $x_t$ and $Y_t$ (Line 2). Then, based on $Y_0, ...Y_t$, the future joint obstacle states $\hat{Y}_{t+1|T}, ..., \hat{Y}_{T|t}$ are predicted using LSTMs (Line 3). Additionally, based on $x_t$ and $D_{cal}^2$, the posterior collision risk can be calculated through (8) (Line 4). After initialization (Line 5), IRA jointly optimizes risk allocation and trajectory through iteration (Line 6-12). Specifically, in each iteration, IRA first computes the optimal control $u_{t:T-1}^n$, state $x_{t+1:T}^n$, and cost $J^*(\alpha_{t+1:T}^n)$ for the current iteration based on the risk allocation $\alpha_{t+1:T}^n$ obtained from the previous iteration (Line 7). The active and inactive constraint sets $\mathcal{I}_{act}, \mathcal{I}_{ina}$ are determined based on the optimal state $x_{t+1:T}^n$ (Line 8). And then, by sequentially applying the inactive constraint tightening (15) (Line 9) and the redundant risk reallocation (17) (Line 10), the updated risk allocation $\alpha_{t+1:T}^{n+1}$ is obtained. Finally, if the convergence condition is satisfied, the the optimal control $u_{t:T-1}^{n-1}$ and the risk allocation $u_{t+1:T}^{n-1}$ are output; otherwise, the next iteration begins (Line 12). Finally, the convergence of Algorithm 5 and the overall risk guarantee are established in Theorems 5.3 and 5.4, whose proofs are provided in Appendices B.6 and B.7, respectively.

**Theorem 5.3** *(convergence guarantee) Assume that $x_{t+1:T}^0, u_{t:T-1}^0$ are feasible in problem (10) with risk allocation $\alpha_{t+1:T}^0$. If the sets $\mathcal{X}, \mathcal{U}$ are bounded and the objective function $J(x_{t+1:T}, u_{t:T-1})$ is continuous, then the sequence of the optimal objective $\{J^*(\alpha_{t+1:T}^n)\}_{n\in\mathbb{N}}$ converges to a finite limit.*

**Theorem 5.4** *(overall risk guarantee for entire trajectory) Given an overall risk tolerance $\alpha$, if the posterior risk $\beta_{1:t}$ is calculated through Lemma 4.2, the risk $\alpha_{t+1:T}$ is allocated through ARA (11) or IRA (12), the planned state $x_{t+1:T}^*$ is a feasible solution of the TO problem (10) with $\alpha_{t+1:T}$, then the entire trajectory at time $t$ satisfies the overall risk guarantee $\mathbb{P}\{\bigcap_{\tau=1}^T \{C(x_\tau^*, Y_\tau) \geq 0\}\} \geq 1 - \alpha$.*

**Remark 5.5** *One may notice that the calculation of $\beta_\tau$ in Lemma 4.2 is similar to the computation of $\widetilde{\alpha}_\tau^n$ in Lemma 5.2. This observation is correct. The key difference between the two lies in that the former utilizes the dataset $D_{cal}^2$ independent with $D_{cal}^1$ to achieve the coverage guarantee for $\beta_\tau$. By contrast, as a step in solving (11), the latter does not need to consider the coverage guarantee and thus directly uses $D_{cal}^1$. The use of different datasets results in the former providing probabilistic guarantee, while the latter achieves deterministic guarantee ($\underline{\alpha}_\tau^n \leq \alpha_\tau^n$).*

---

**Algorithm 1** Fb-CP using IRA at time $t$

---

1: **Input:** $\alpha$, $\alpha_{t:T}$, $\beta_{0:t-1}$, $\epsilon$, $\eta$, $D_{cal}^1$, $D_{cal}^2$
2: Observe the system state $x_t$ and joint obstacle states $Y_t$
3: $\hat{Y}_{t+1|t}, ..., \hat{Y}_{T|t} \leftarrow$ Trajectory prediction using LSTMs based on $Y_0, ..., Y_t$
4: $\beta_t \leftarrow$ Posterior probability calculation (8) {Using $x_t$ and $D_{cal}^2$}
5: $J^*(\alpha_{t+1:T}^{-1}) \leftarrow \infty$, $\alpha_{t+1:T}^0 \leftarrow \alpha_{t+1:T}$, $n \leftarrow 0$ {Initialization of IRA}
6: **repeat**
7: $\quad J^*(\alpha_{t+1:T}^n)$, $x_{t+1:T}^n$, $u_{t:T-1}^n \leftarrow$ Solve the lower-stage problem (10) with $\alpha_{t+1:T}^n$
8: $\quad \mathcal{I}_{act}$, $\mathcal{I}_{ina}$, $N_{act} \leftarrow$ Identification of active and inactive constraints
9: $\quad \widetilde{\alpha}_{t+1:T}^n \leftarrow$ Transitional risk allocation calculation (15)
10: $\quad \alpha_{t+1:T}^{n+1} \leftarrow$ New risk allocation calculation (17)
11: $\quad n \leftarrow n + 1$
12: **until** $|J^*(\alpha_{t+1:T}^{n-1}) - J^*(\alpha_{t+1:T}^{n-2})| < \epsilon$
13: **Output:** $\beta_{0:t}$, $u_{t:T-1}^{n-1}$, $\alpha_{t+1:T} = \alpha_{t+1:T}^{n-1}$

---

## 6 Experiments

All the experiments are performed on a personal computer with 2.10 GHz Inter Core i7-13700 CPU and 32 GB RAM. We conduct 1,000 Monte Carlo experiments on a kinematic vehicle model Lekeufack et al. [2024], a 3D linear quadrotor model Dixit et al. [2023], a dynamic bicycle model Hakobyan and Yang [2021] and the Stanford Drone Dataset Robicquet et al. [2016] to compare our method with the state-of-art methods as follows [2].

(i) Conformal Control (CC) proposed in Lekeufack et al. [2024].
(ii) ACI for Motion Planning (ACI-MP) proposed in Dixit et al. [2023].
(iii) Recursively Feasible MPC using CP (RF-CP) proposed in Stamouli et al. [2024]
(iv) Sequential CP (S-CP) proposed in Lindemann et al. [2023].
(v) Fb-CP with ARA (Fb-CP-ARA): The method based on Fb-CP using average risk allocation.
(vi) Fb-CP with IRA (Fb-CP-IRA): The method based on Fb-CP using iterative risk allocation.

In this section, we present the main experimental results, while the full set of results can be found in Appendix C. Figure 2 shows the simulation results from one of the 1,000 simulations using the 2D vehicle model. At $t = 0$, the vehicle performs the first TO. For Fb-CP-IRA, IRA allows for flexible allocation of the risks across future times. Therefore, by assigning more risk to $\tau = 9$, which leads to a compact prediction region, a trajectory passing between Obstacles 2 and 3 is obtained. However, with the fixed risk allocation at $t = 0$, S-CP and Fb-CP-ARA can only optimize the trajectory based on fixed prediction regions. Consequently, they can only navigate around to pass between Obstacles 1 and 2. Note that at $t = 0$, no deterministic vehicle position is available for posterior probability calculation. Thus Fb-CP-ARA degenerates into S-CP, resulting in both methods obtaining essentially the same trajectory. As time progresses, more and more vehicle positions become available. For Fb-CP-ARA, $\beta_{1:3}$ can be computed at $t = 3$ and is with high probability less than $\alpha_{1:3}$, as outlined in Corollary A.2. The reduction from $\alpha_{1:3}$ to $\beta_{1:3}$ leads to an increased allowable risk for future times, corresponding to a narrowing in the prediction regions.Therefore, compared with S-CP, Fb-CP-ARA generates a less conservative trajectory. Similarly, Fb-CP-IRA also leverages $\beta_{1:3}$ to increase the total allocable risk, thereby further enhancing the flexibility in allocating risks. As shown in Figure 2, the trajectory obtained by Fb-CP-IRA at $t = 3$ exhibits reduced conservativeness compared with the trajectory obtained at $t = 0$.

Table 1 summarizes the average cost, computation time, and collision avoidance rate of 1,000 simulations using the quadrotor model with different methods. As shown in Table 1, the Fb-CP-ARA reduces the cost by an average of 11.34% compared with S-CP thanks to the feedback information of posterior probabilities, with a negligible additional computational burden. Furthermore, by flexibly allocating the additional allowable risk provided by posterior probabilities, Fb-CP-IRA achieves an 58.50% reduction in average cost compared with S-CP. However, since IRA needs to solve the TO problem (10) iteratively, the average computation time increases significantly. Additionally, since CC and ACI-MP fail to fully utilize the information in the calibration dataset, they incur higher costs, which are 184% and 296% higher than those of Fb-CP-IRA, respectively. Particularly for CC, it directly controls the collision avoidance rate by adjusting the weight of the collision penalty term

---

[2]`https://github.com/DOCU-Lab/Feedback-based_Conformal_Prediction`

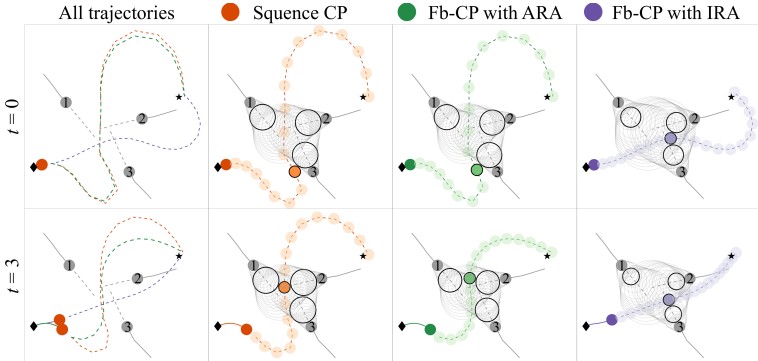

Figure 2: Trajectories of the vehicle with different TO methods. (Numbers on the circles denote the indices of obstacles. The diamond and pentagon symbols represent the initial and target points of the vehicle, respectively. The translucent circles represent the planned positions of the vehicle and the prediction regions for further time. In particular, the colored and transparent circles with black edges denote the planned positions and the prediction regions for $\tau = 9$, respectively.)

in the objective function, which results in a higher average cost. However, it should be noted that, in practice, CC and ACI-MP are better suited for scenarios where the test data exhibit distribution shift, rather than the setup considered in our work. For RF-CP, thanks to the proposed normalized nonconformity score, its average cost is comparable to that of Fb-CP-ARA, but it remains 85.8% higher than Fb-CP-IRA. However, the normalized nonconformity score introduces mixed-integer variables into the TO problem, significantly increasing the computation time. Specifically, the average computation time of RF-CP is more than an order of magnitude higher than that of Fb-CP-IRA.

Additionally, we have investigated the impact of using prior versus posterior probabilities on the prediction regions by analyzing the prediction region radius at different time $t$ and $\tau$ in Appendix H. Furthermore, we have also extended Fb-CP to handle distribution shift and empirically demonstrate its effectiveness. Detailed methodology and corresponding experiments can be found in Appendix G.

Table 1: Average cost, computation time, and collision avoidance rate using the quadrotor model with different methods ($\eta$ is the learning rate of CC).

| | | CC | | ACI-MP | RF-CP | S-CP | Fb-CP | |
| --- | --- | --- | --- | --- | --- | --- | --- | --- |
| | | | | | | | ARA | IRA |
| Average cost | $\eta = 1000$ | 59.25 | $\alpha = 0.05$ | 17.970 | 15.794 | 17.321 | 15.356 | 7.189 |
| | $\eta = 500$ | 47.50 | $\alpha = 0.10$ | 17.263 | 14.378 | 16.17 | 14.228 | 6.798 |
| | $\eta = 100$ | 22.46 | $\alpha = 0.15$ | 16.096 | 11.922 | 14.83 | 12.354 | 6.191 |
| | $\eta = 50$ | 21.34 | $\alpha = 0.20$ | 15.310 | 10.032 | 13.217 | 10.22 | 5.398 |
| Average computation time | $\eta = 1000$ | 0.019 | $\alpha = 0.05$ | 0.022 | 0.487 | 0.022 | 0.027 | 0.038 |
| | $\eta = 500$ | 0.019 | $\alpha = 0.10$ | 0.026 | 0.494 | 0.020 | 0.021 | 0.039 |
| | $\eta = 100$ | 0.021 | $\alpha = 0.15$ | 0.021 | 0.545 | 0.021 | 0.020 | 0.037 |
| | $\eta = 50$ | 0.022 | $\alpha = 0.20$ | 0.022 | 0.500 | 0.020 | 0.019 | 0.036 |
| Collision avoidance rate | $\eta = 1000$ | 97.0% | $\alpha = 0.05$ | 98.6% | 98.7% | 98.8% | 98.2% | 96.3% |
| | $\eta = 500$ | 92.8% | $\alpha = 0.10$ | 93.3% | 96.9% | 93.5% | 94.6% | 94.1% |
| | $\eta = 100$ | 82.5% | $\alpha = 0.15$ | 91.5% | 92.4% | 92.0% | 90.2% | 91.9% |
| | $\eta = 50$ | 79.1% | $\alpha = 0.20$ | 87.9% | 90.0% | 88.2% | 86.7% | 88.2% |

## 7 Conclusion and Limitations

In this paper, we proposed a Fb-CP framework for shrinking-horizon TO with a joint risk constraint in uncertain environments. This method enables the feedback of the information in the realized trajectory from the decision-making to the CP, guiding the closed-loop adjustments of the prediction regions. The proposed adjustment rule balances both performance and safety, offering provable performance and coverage guarantees. The limitations are discussed in detail in Appendix L.

## Acknowledgements

This work was supported in part by the National Natural Science Foundation of China under Grants 62473256 and 62103264.

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

# A  Auxiliary Results

## A.1  Conformal Prediction Lemma

**Lemma A.1** *[Lemma 1 in Tibshirani et al. [2019]]* (coverage guarantee) If $R^{(0)}, R^{(1)}, ..., R^{(N)}$ are $N + 1$ exchangeable random variables, then for a failure probability $\alpha \in (0, 1)$, it holds that

$$\mathbb{P}\left\{R^{(0)} \leq Quantile_{1-\alpha}(R^{(1)}, ..., R^{(N)}, \infty)\right\} \geq 1 - \alpha \tag{19}$$

where the function $Quantile_{1-\alpha}(R^{(1)}, ..., R^{(N)}, \infty)$ denotes the level $1-\alpha$ quantile of the empirical distribution of the values $R^{(1)}, ..., R^{(N)}, \infty$ as follows.

$$Quantile_{1-\alpha}(R^{(1)}, ..., R^{(N)}, \infty) = \inf\{z : \mathbb{P}\{Z \leq z\} \geq 1 - \alpha\}, \tag{20a}$$

$$Z \sim \left(\textstyle\sum_{i=1}^{N} \delta_{R^{(i)}} + \delta_{\infty}\right)/(N + 1) \tag{20b}$$

where $\delta_{R^{(i)}}$ and $\delta_{\infty}$ denote the Dirac delta function at $R^{(i)}$ and $\infty$, respectively.

## A.2  Upper bound of the expectation of posterior risk

**Corollary A.2** *(upper bound of $\beta_\tau$) Suppose that $\delta \in (0, 1)$ and $K > (-\ln \delta)/(2\alpha_\tau^2)$, we have*

$$\mathbb{P}\left\{\mathbb{E}(\beta_\tau) \leq \left(1 + L\left(\alpha_\tau + \sqrt{-\ln \delta/(2K)}\right)\right)/(1 + L)\right\} \geq 1 - \delta \tag{21}$$

*Furthermore if $K, L \to \infty$, then $\mathbb{E}(\beta_\tau) \leq \alpha_\tau$ holds with probability one.*

**Remark A.3** *Corollary A.2 provides a performance guarantee for Fb-CP, i.e. Fb-CP performs at least as well as the sequential method Lindemann et al. [2023] with high probability. Furthermore, the experiments in Section 6 demonstrate that the proposed method performs significantly better in practice. This is attributed to the conservatism of the inequality (38) in the proof of Corollary A.2. Since $c(x_\tau^*, Y_\tau)$ contains the information provided by $c(\cdot)$ (e.g. the shape of the robot and obstacles) and $x_\tau^*$, it typically occurs that $\mathbb{P}\{c(x_\tau^*, Y_\tau) < 0\} \ll \mathbb{P}\{\|Y_\tau - \hat{Y}_{\tau|t}\| > C_{\tau|t}^{1-\alpha_\tau}\}$.*

# B  Proofs

## B.1  Proof of Lemma 4.1

According to Assumptions 3.1 as well as the calculation of $C_{\tau|t}^{1-\alpha_\tau}$ (6b), the $(1 - \alpha_\tau)$-coverage guarantee of the prediction (6a) is obtained through Lemma $A.1$. Note that the function $c$ is $L$-Lipschitz continuous, the following inequality is obtained.

$$\|c(x_\tau, Y_\tau) - c(x_\tau, \hat{Y}_{\tau|t})\| \leq L\|Y_\tau - \hat{Y}_{\tau|t}\| \implies c(x_\tau, Y_\tau) \geq c(x_\tau, \hat{Y}_{\tau|t}) - L\|Y_\tau - \hat{Y}_{\tau|t}\| \tag{22}$$

If the constraint $c(x_\tau, \hat{Y}_{\tau|t}) \geq LC_{\tau|t}^{1-\alpha_\tau}$ is satisfied, we have the following inequality.

$$c(x_\tau, Y_\tau) \geq L(C_{\tau|t}^{1-\alpha} - \|Y_\tau - \hat{Y}_{\tau|t}\|) \tag{23}$$

According to the $(1 - \alpha_\tau)$-coverage guarantee (6a) $\mathbb{P}\{C_{\tau|t}^{1-\alpha} - \|Y_\tau - \hat{Y}_{\tau|t}\| \geq 0\} \geq 1 - \alpha_\tau$, the lemma is proven.

## B.2  Proof of Lemma 4.2

According to Assumption 3.1, we know that all trajectories $Y^{(i)}$ are generated by sampling their initial state $Y_0^{(i)}$ from the same distribution $\mathbb{D}$ and then evolving using a common dynamics (the ground truth dynamics). As a result, the random variables $\omega_\tau, \omega_\tau^{(1)}, ..., \omega_\tau^{(K+L)}$ are i.i.d.. Note that $x_\tau^*$ is the true state of the system at time $\tau$, thus $x_\tau^*$ is fixed and independent of $\omega_\tau, \omega_\tau^{(K+1)}, ..., \omega_\tau^{(K+L)}$. Therefore, the random variables $S_\tau, S_\tau^{(K+1)}, ..., S_\tau^{(K+L)}$ are exchangeable.

Without loss of generality, we assume that the dataset $\{-S_\tau^{(K+i)} : i = 1, ..., L\}$ are sorted in non-decreasing order. We first assume that $-S_\tau^{(K+1)} \leq 0$, and then we define the maximum index $\ell$ that makes $-S_\tau^{(K+\ell)} \leq 0$ hold as follows.

$$\ell = \max_{l=1,...,L} \quad l$$
$$s.t. \quad -S_\tau^{(K+\ell)} \leq 0 \tag{24}$$

Then the posterior satisfaction probability can be computed below.

$$\mathbb{P}\{c(x_\tau^*, Y_\tau) \geq 0\} = \mathbb{P}\{-S_\tau \leq 0\} \geq \mathbb{P}\{-S_\tau \leq -S_\tau^{(K+\ell)}\} \tag{25}$$

It is assumed that there are $t$ terms in $\{-S_\tau^{(K+i)} : i = 1, ..., L\}$ identical to $-S_\tau^{(K+\ell)}$, i.e.

$$-S_\tau^{(K+\ell-t)} < -S_\tau^{(K+\ell-t+1)} = ... = -S_\tau^{(K+\ell)} \leq 0 < -S_\tau^{(K+\ell+1)} \tag{26}$$

Then $-S_\tau^{(K+\ell)}$ can be equivalently reformulated as follows.

$$-S_\tau^{(K+\ell)} = Quantile_\beta(-S_\tau^{(K+1)}, ..., -S_\tau^{(K+L)}, \infty), \quad \forall \beta \in \left( \frac{\ell-t}{1+L}, \frac{\ell}{1+L} \right] \tag{27}$$

Combining (25) and (27) we have

$$\mathbb{P}\{c(x_\tau^*, Y_\tau) \geq 0\} \geq \mathbb{P}\{-S_\tau \leq Quantile_\beta(-S_\tau^{(K+1)}, ..., -S_\tau^{(K+L)}, \infty)\} \tag{28}$$

Note that the random variables $S_\tau, S_\tau^{(K+1)}, ..., S_\tau^{(K+L)}$ are exchangeable and $\beta \in \left( \frac{\ell-t}{1+L}, \frac{\ell}{1+L} \right] \subset (0, 1)$, and thus we can apply Lemma A.1 and obtain

$$\mathbb{P}\{c(x_\tau^*, Y_\tau) \geq 0\} \geq \beta, \quad \forall \beta \in \left( \frac{\ell-t}{1+L}, \frac{\ell}{1+L} \right] \tag{29}$$

Therefore, the upper bound of the posterior violation probability can be computed by

$$\mathbb{P}\{c(x_\tau^*, Y_\tau) < 0\} \leq 1 - \beta, \quad \forall \beta \in \left( \frac{\ell-t}{1+L}, \frac{\ell}{1+L} \right] \tag{30}$$

To minimize this upper bound, we take the maximum value of $\beta$ and (30) becomes (31).

$$\mathbb{P}\{c(x_\tau^*, Y_\tau) < 0\} \leq 1 - \frac{\ell}{1+L} \tag{31}$$

According to the definition of $\ell$ (24), we can compute $\ell$ as follows.

$$\ell = \sum_{i=1}^{L} \mathbb{I}\left(S_\tau^{(K+i)} \geq 0\right) = L - \sum_{i=1}^{L} \mathbb{I}\left(S_\tau^{(K+i)} < 0\right) \tag{32}$$

Combining (31) and (32), we have

$$\mathbb{P}\{c(x_\tau^*, Y_\tau) < 0\} \leq \frac{1 + \sum_{i=1}^{L} \mathbb{I}\left(S_\tau^{(K+i)} < 0\right)}{1+L} \tag{33}$$

Finally, we consider the scenario in which $-S_\tau^{(K+1)} > 0$, which means $S_\tau^{(K+i)} < 0 \; \forall i = 1, ..., L$. Then the inequality (33) is simplified as follows.

$$\mathbb{P}\{c(x_\tau^*, Y_\tau) < 0\} \leq 1 \tag{34}$$

which is always true. Thus, the Lemma is proven.

## B.3 Proof of Corollary A.2

Taking expectations on both sides of Equation (8), we can obtain

$$\mathbb{E}(\beta_\tau) = \frac{1 + L\mathbb{P}\{S_\tau^{(K+i)} < 0\}}{1 + L} = \frac{1 + L\mathbb{P}\{S_\tau < 0\}}{1 + L} = \frac{1 + L\mathbb{P}\{c(x_\tau^*, Y_\tau) < 0\}}{1 + L} \tag{35}$$

The second equality holds because $S_\tau, S_\tau^{K+1}, ..., S_\tau^{K+L}$ are exchangeable. Note that the function $c()$ is $L$-Lipschitz continuous and $x_\tau^*$ is a feasible solution of problem (3) with the reformulated constraint through Lemma 4.1, and the following inequality can be derived in the same manner as inequalities (22) and (23) in the Proof of Lemma 4.1 (Appendix B.1).

$$c(x_\tau^*, Y_\tau) \geq L(C_{\tau|t}^{1-\alpha_\tau} - \|Y_\tau - \hat{Y}_{\tau|t}\|) \tag{36}$$

Based on (36), we can obtain

$$c(x_\tau^*, Y_\tau) < 0 \Rightarrow \|Y_\tau - \hat{Y}_{\tau|t}\| > C_{\tau|t}^{1-\alpha_\tau} \tag{37}$$

And the following inequality is derived.

$$\mathbb{P}\{c(x_\tau^*, Y_\tau) < 0\} \leq \mathbb{P}\{\|Y_\tau - \hat{Y}_{\tau|t}\| > C_{\tau|t}^{1-\alpha_\tau}\} \tag{38}$$

Combining (35) and (38), we have

$$\mathbb{E}(\beta_\tau) \leq \frac{1 + L\mathbb{P}\{\|Y_\tau - \hat{Y}_{\tau|t}\| > C_{\tau|t}^{1-\alpha_\tau}\}}{1 + L} \tag{39}$$

For $\alpha_\tau, \delta \in (0,1)$ and $K > (-\ln \delta)/(2\alpha_\tau^2)$, we can apply [Vovk [2012], Proposition 2a] so that

$$\mathbb{P}\left\{\mathbb{P}\left\{\|Y_\tau - \hat{Y}_{\tau|t}\| \leq C_{\tau|t}^{1-\alpha_\tau}\right\} \geq 1 - \left(\alpha_\tau + \sqrt{-\ln \delta/(2K)}\right)\right\} \geq 1 - \delta \tag{40}$$

which can be equivalently transformed into the following expression.

$$\mathbb{P}\left\{\mathbb{P}\left\{\|Y_\tau - \hat{Y}_{\tau|t}\| > C_{\tau|t}^{1-\alpha_\tau}\right\} \leq \alpha_\tau + \sqrt{-\ln \delta/(2K)}\right\} \geq 1 - \delta \tag{41}$$

Combining (39) and (41), we can finally obtain the inequality (21).

When $K, L \to \infty$, we can further assume that $L \geq 1/\delta$ and $K \geq \max\{(-\ln \delta)/(2\alpha_\tau^2), 1/\delta\}$. Note that for a fixed $\alpha_\tau$, we can always find a small enough positive $\delta$ such that $\alpha_\tau + \sqrt{(-\ln \delta)/(2K)} < \alpha_\tau + \sqrt{(-\delta \ln \delta)/2} < 1$. Therefore for a small enough positive $\delta$ we have

$$\mathbb{P}\left\{\mathbb{E}(\beta_\tau) \leq \frac{\delta + \alpha_\tau + \sqrt{-\delta \ln \delta/2}}{\delta + 1}\right\}$$
$$\geq \mathbb{P}\left\{\mathbb{E}(\beta_\tau) \leq \frac{1 + L\left(\alpha_\tau + \sqrt{-\ln \delta/(2K)}\right)}{1 + L}\right\} \geq 1 - \delta \tag{42}$$

Let $\delta \to 0^+$, we finally obtain that $\mathbb{E}(\beta_\tau) \leq \alpha_\tau$ holds with probability one.

## B.4 Proof of Lemma 5.1

Let $\alpha_{t+1:T}^1$ and $\alpha_{t+1:T}^2$ be two risk allocations at time $t$. Based on the definition of $C_{\tau|t}^{1-\alpha_\tau}$ (6b), $C_{\tau|t}^{1-\alpha_\tau}$ is non-increasing with respect to $\alpha_\tau$ for fixed $D_{cal}^1$. Therefore, if $\alpha_\tau^1 \leq \alpha_\tau^2$, $\forall \tau = t+1, ..., T$, then $C_{\tau|t}^{1-\alpha_\tau^1} \geq C_{\tau|t}^{1-\alpha_\tau^2}$ which further leads to $\mathcal{R}_t(\alpha_{t+1:T}^1) \subseteq \mathcal{R}_t(\alpha_{t+1:T}^2)$. Since $J^*(\alpha_{t+1:T})$ is the minimum of the objective problem (10) with the feasible region $\mathcal{R}_t(\alpha_{t+1:T})$, $J^*(\alpha_{t+1:T}^1) \geq J^*(\alpha_{t+1:T}^2)$ can be obtained and the lemma is proven.

## B.5 Proof of Lemma 5.2

The computation of the lower bound is analogous to the calculation of $\beta_\tau$ in Lemma 4.1, except that Lemma A.1 is not required to obtain coverage guarantees. Therefore, the computation is based on $D_{cal}^1$. Without loss of generality, we assume that the dataset $\{R_{\tau|t}^{(i)} : i = 1, ..., K\}$ is sorted in non-decreasing order. Note that $x_{t+1:T}^n$ is feasible for the problem (10) with $\alpha_{t+1:T}^n$ and $\tau \in \mathcal{I}_{ina}$, the inequality $c(x_\tau^n, \hat{Y}_{\tau|t}) > LC_{\tau|t}^{1-\alpha_\tau^n} = LQuantile_{1-\alpha_\tau^n}(R_{\tau|t}^{(1)}, ..., R_{\tau|t}^{(K)}, \infty)$ holds true. Since $\alpha_\tau^n < 1$, it follows that $c(x_\tau^n, \hat{Y}_{\tau|t}) > R_{\tau|t}^{(1)}$. Therefore, we define the maximum index $\mathcal{K}$ that makes $c(x_\tau^n, \hat{Y}_{\tau|t}) \geq LR_{\tau|t}^{(\mathcal{K})}$ hold as follows.

$$\mathcal{K} = \max_{k=1,...,K} \quad k \tag{43}$$
$$s.t. \quad c(x_\tau^n, \hat{Y}_{\tau|t}) \geq LR_{\tau|t}^{(k)}$$

It is assumed that there are $t$ terms in $\{R_{\tau|t}^{(i)} : i = 1, ..., K\}$ identical to $R_{\tau|t}^{\mathcal{K}}$, and thus we can obtain

$$R_{\tau|t}^{(\mathcal{K}-t)} < R_{\tau|t}^{(\mathcal{K}-t+1)} = ... = R_{\tau|t}^{(\mathcal{K})} \leq c(x_\tau^n, \hat{Y}_{\tau|t})/L < R_{\tau|t}^{(\mathcal{K}+1)} \tag{44}$$

We aim to determine the maximum value of $C_{\tau|t}^{1-\widetilde{\alpha}_\tau^n}$ (the minimum value of $\widetilde{\alpha}_\tau^n$) while satisfying $C_{\tau|t}^{1-\widetilde{\alpha}_\tau^n} \leq c(x_\tau^n, \hat{Y}_{\tau|t})/L$, which is equivalent to $C_{\tau|t}^{1-\widetilde{\alpha}_\tau^n} \leq R_{\tau|t}^{\mathcal{K}}$ because $C_{\tau|t}^{1-\widetilde{\alpha}_\tau^n}$ can only take values at a finite number of discrete points $R_{\tau|t}^{(1)}, ..., R_{\tau|t}^{(K)}, \infty$. Furthermore, $R_{\tau|t}^{\mathcal{K}}$ can be equivalently reformulated as follows.

$$R_{\tau|t}^{\mathcal{K}} = Quantile_\beta(R_{\tau|t}^{(1)}, ..., R_{\tau|t}^{(K)}, \infty) = C_{\tau|t}^\beta, \quad \forall \beta \in \left(\frac{\mathcal{K}-t}{1+K}, \frac{\mathcal{K}}{1+K}\right] \tag{45}$$

Therefore, the constraint $C_{\tau|t}^{1-\widetilde{\alpha}_\tau^n} \leq R_{\tau|t}^{\mathcal{K}}$ is equivalent to the following expression.

$$C_{\tau|t}^{1-\widetilde{\alpha}_\tau^n} \leq C_{\tau|t}^\beta, \quad \exists \beta \in \left(\frac{\mathcal{K}-t}{1+K}, \frac{\mathcal{K}}{1+K}\right] \tag{46}$$

Note that $C_{\tau|t}^\beta$ is non-decreasing with respect to $\beta$ for fixed $D_{cal}^1$. Constraint (46) is further reformulated as follows.

$$\widetilde{\alpha}_\tau^n \geq 1 - \frac{\mathcal{K}}{1+K} \tag{47}$$

According to the definition of $\mathcal{K}$ (43), we can compute $\mathcal{K}$ in (48).

$$\mathcal{K} = \sum_{i=1}^K \mathbb{I}\left(c(x_\tau^n, \hat{Y}_{\tau|t}) \geq LR_{\tau|t}^{(i)}\right) = K - \sum_{i=1}^K \mathbb{I}\left(c(x_\tau^n, \hat{Y}_{\tau|t}) < LR_{\tau|t}^{(i)}\right) \tag{48}$$

Combining (47) and (48), the lower bound of $\widetilde{\alpha}_\tau^n$ is computed as follows.

$$\underline{\alpha}_\tau^n = \frac{1 + \sum_{i=1}^K \mathbb{I}\left(c(x_\tau^n, \hat{Y}_{\tau|t}) < LR_{\tau|t}^{(i)}\right)}{1+K} \tag{49}$$

We note that $\underline{\alpha}_\tau^n$ is the lower bound of $\widetilde{\alpha}_\tau^n$ that ensures the constraint $c(x_\tau^n, \hat{Y}_{\tau|t}) \geq LC_{\tau|t}^{1-\widetilde{\alpha}_\tau^n}$. Furthermore, since $x_{t+1:T}^n$ is feasible for the problem (10) with $\alpha_{t+1:T}^n$, the constraint $c(x_\tau^n, \hat{Y}_{\tau|t}) \geq LC_{\tau|t}^{1-\alpha_\tau^n}$ is satisfied. Therefore, $\underline{\alpha}_\tau^n \leq \alpha_\tau^n$ is naturally obtained. Thus the Lemma is proven.

## B.6 Proof of Theorem 5.3

The proof adapts elements of the proof from Zymler et al. [2013]. If $x_{t+1:T}^0, u_{t:T-1}^0$ is a feasible solution for the risk allocation $\alpha_{t+1:T}^0$, the update law of $\alpha_{t+1:T}$ guarantees that the sequence of the optimal objective values $\{J^*(\alpha_{t+1:T}^n)\}_{n\in\mathbb{N}}$ is monotonically decreasing, as previously mentioned. Since the sets $\mathcal{X}$ and $\mathcal{U}$ are bounded, $x_{t+1:T}$ and $u_{t:T-1}$ are bounded. Because the objective function $J(x_{t+1:T}, u_{t:T-1})$ is continuous, the boundedness of $x_{t+1:T}, u_{t:T-1}$ and the monotonicity of the optimal objective value sequence imply that $\{J^*(\alpha_{t+1:T}^n)\}_{n\in\mathbb{N}}$ converges to a finite limit.

## B.7 Proof of Theorem 5.4

At each time $t$, the realized state $x_{1:t}^*$ is available. According to Lemma 4.2, we know that

$$\mathbb{P}\{c(x_\tau^*, Y_\tau) < 0\} \leq \beta_\tau, \quad \forall \tau = 1, ..., t \tag{50}$$

For the planned state $x_{t+1:T}^*$, the corresponding risk allocations $\alpha_{t+1:T}$ are obtained either via ARA (11) or IRA (12). In both cases, they satisfy

$$\sum_{\tau=t+1}^{T} \alpha_\tau \leq \alpha - \sum_{\tau=1} \beta_\tau \tag{51}$$

We note that if the planned state $x_{t+1:T}^*$ is a feasible solution of the TO problem (10) with $\alpha_{t+1:T}$, the following inequality can be obtained through Lemma 4.1.

$$\mathbb{P}\{c(x_\tau^*, Y_\tau) \geq 0\} \geq 1 - \alpha_\tau, \quad \forall \tau = t+1, ..., T \tag{52}$$

Finally, by combining the above three inequalities (50), (51), and (52), together with Boole's inequality, we obtain

$$\mathbb{P}\{\bigcap_{\tau=1}^{T} \{C(x_\tau^*, Y_\tau) \geq 0\}\} \geq 1 - \alpha \tag{53}$$

which means that the overall risk of the entire trajectory at time $t$, $x_{1:T}^*$ is below the risk tolerance $\alpha$. Thus, Theorem 5.4 is proven.

## C   Experiment details and additional results

### C.1   Simulation for a kinematic vehicle model

We examine the kinematic vehicle model Lekeufack et al. [2024] with the following nonlinear dynamics.

$$\begin{bmatrix} p_{x,t+1} \\ p_{y,t+1} \\ \theta_{t+1} \\ v_{t+1} \end{bmatrix} = \begin{bmatrix} p_{x,t} + \Delta v_t \cos \theta_t \\ p_{y,t} + \Delta v_t \sin \theta_t \\ \theta_t + \Delta \frac{v_t}{l} \tan \phi_t \\ v_t + \Delta a_t \end{bmatrix} \tag{54}$$

where $p_t := (p_{x,t}, p_{y,t})$, $\theta_t$, $v_t$ are the position, orientation, and velocity of the vehicle, respectively. $l := 0.2$ is the length, and $\Delta = 0.125$ is the sampling time. The system inputs are the steering angle $\phi_t \in [-\pi/6, \pi/6]$ and the acceleration $a_t \in [-5, 5]$. The total time is set to $T = 20$. The objective is to reach the vicinity of the target point while avoiding collisions with obstacles. Formally, the objective function is defined as $J = \sum_{\tau=t}^{T-1} 100\phi_\tau^2 + a_\tau^2$ to minimize energy consumption and the constraint $\|p_T - p_{tar}\|_2 \leq 0.2$ is incorporated into (9) to ensure the vehicle reaches the target point, where $p_{tar}$ is the target point. The constraint function for collision avoidance is as follows.

$$c(p_\tau, Y_\tau) = \min_{j=1,...,M} \|p_\tau, Y_{\tau,j}\|_2 - r_r - r_o - r_s \tag{55}$$

where $r_r$ and $r_o$ are the inflation radius of the vehicle and obstacle, respectively. $r_s$ is the safety margin. The interior-point method-based solver IPOPT (v3.12.9) was used to solve the TO problem (9). Similar to Lindemann et al. [2023], we consider $M = 3$ obstacles, with their trajectories generated by TrajNet++ Kothari et al. [2021] which is publicly available at https://github.com/vita-epfl/trajnetplusplusbaselines and is licensed under the MIT License. We generate 13,000 joint obstacle trajectories and randomly divide them into training $D_{train}$, calibration $D_{cal}$, and test $D_{test}$ datasets with the set sizes 2,000, 10,000, and 1,000, respectively. We train an LSTM Alahi et al. [2016] using $D_{train}$ as the trajectory predictor. For the proposed Fb-CP, $D_{cal}$ is further divided into $D_{cal}^1$ and $D_{cal}^2$ with sizes $|D_{cal}^1| = 2,000$ and $|D_{cal}^2| = 8,000$. We conduct 1,000 Monte Carlo simulations using $D_{test}$. As we discussed in Section 6, the methods S-CP, Fb-CP-ARA, and Fb-CP-IRA are analyzed.

Table 2 shows the average cost, average computation time, and collision avoidance rate of 1,000 simulations using the kinematic vehicle model with different methods. We collect the simulation data under different total risk tolerances $\alpha = 0.05, 0.10, 0.15, 0.20$. On one hand, with an increase in

total risk tolerance, the average cost of all methods decreases. On the other hand, benefiting from the feedback information of posterior probabilities, the average cost of Fb-CP-ARA shows a reduction of 7.21% to 11.26% compared to S-CP. Furthermore, by flexibly allocating the allowable risk provided by posterior probabilities, the average cost of Fb-CP with IRA exhibits a significant reduction compared with S-CP. Additionally, the increase in total risk tolerance provides greater flexibility in the risk allocation of Fb-CP-IRA, resulting in a significant reduction in its average cost. As mentioned in Section 6, the calculation of posterior probabilities does not incur additional computational burden. Therefore, the average computation time of Fb-CP-ARA is essentially comparable to that of S-CP. The collision rates of all methods do not exceed their corresponding total risk tolerances.

Table 2: Average cost, computation time, and collision avoidance rate using the kinematic vehicle model with different methods.

| | | Sequential CP | Fb-CP | |
| | | | with ARA | with IRA |
| --- | --- | --- | --- | --- |
| | $\alpha = 0.05$ | 22.20 | 20.46 | 4.77 |
| Average cost | $\alpha = 0.10$ | 20.24 | 18.78 | 3.52 |
| | $\alpha = 0.15$ | 19.22 | 17.35 | 3.18 |
| | $\alpha = 0.20$ | 17.05 | 15.13 | 2.89 |
| | $\alpha = 0.05$ | 0.111 | 0.100 | 0.128 |
| Average computation time | $\alpha = 0.10$ | 0.093 | 0.087 | 0.126 |
| | $\alpha = 0.15$ | 0.078 | 0.085 | 0.130 |
| | $\alpha = 0.20$ | 0.076 | 0.078 | 0.131 |
| | $\alpha = 0.05$ | 95.4% | 95.7% | 98.4% |
| Collision avoidance rate | $\alpha = 0.10$ | 93.9% | 93.1% | 98.3% |
| | $\alpha = 0.15$ | 90.8% | 89.8% | 97.6% |
| | $\alpha = 0.20$ | 88.4% | 89.1% | 91.2% |

### C.2 Simulation for linear quadrotor model

We examine the quadrotor model Dixit et al. [2023] with the following linear dynamics.

$$
\begin{array}{lll}
\ddot{x} = g\theta & \ddot{y} = -g\phi & \ddot{z} = \frac{1}{m_Q}u_1 \\
\ddot{\phi} = \frac{l_Q}{I_{xx}}u_2 & \ddot{\theta} = \frac{l_Q}{I_{yy}}u_3 & \ddot{\psi} = \frac{l_Q}{I_{zz}}u_4
\end{array}
\tag{56}
$$

where $g = 9.81$ represents the gravitational acceleration, $m_Q = 0.65$ denotes the mass, and $l_Q = 0.23$ is the distance between the quadrotor and the rotor. $I_{xx} = 0.0075$, $I_{yy} = 0.0075$, and $I_{zz} = 0.013$ correspond to the area moments of inertia about the principle axes in the body frame. The states are the position and orientation with the corresponding velocities and rates — $(x, y, z, \dot{x}, \dot{y}, \dot{z}, \phi, \theta, \psi, \dot{\phi}, \dot{\theta}, \dot{\psi}) \in \mathbb{R}^{12}$. The control inputs $u_1, u_2, u_3, u_4$ correspond to the thrust force in the body frame and three moments. The system (56) is discretized using the sampling time $\Delta = 0.125$, and the total time is also set to $T = 20$.

Similar to the experiments based on the kinematic vehicle model in Appendix C.1, the objective is to control the quadrotor to reach the target point $p_{tar}$ while navigating around $M = 3$ moving obstacles. The target point constraint and obstacle avoidance constraints are consistent with those used in the simulation using the kinematic vehicle model. We randomly generate 13,000 obstacle trajectories and assign them as in Appendix C.1. The following state-of-art methods recently proposed in the literature are analyzed through 1,000 Monte Carlo simulations.
(i) Conformal Control (CC) proposed in Lekeufack et al. [2024].
(ii) ACI for Motion Planning (ACI-MP) proposed in Dixit et al. [2023].
(iii) Recursively Feasible MPC using CP (RF-CP) proposed in Stamouli et al. [2024]
(iv) Sequential CP (S-CP) proposed in Lindemann et al. [2023]. Computation of the CP region and TO is performed sequentially.
(v) Fb-CP with ARA (Fb-CP-ARA): The method based on Fb-CP using average risk allocation.
(vi) Fb-CP with IRA (Fb-CP-IRA): The method based on Fb-CP using iterative risk allocation.

Table 3 shows the average cost, average computation time, and collision avoidance rate of 1,000 simulations using the quadrotor model with different methods. For the methods S-CP, Fb-CP-ARA,

Table 3: Average cost, computation time, and collision avoidance rate using the quadrotor model with different methods ($\eta$ is the learning rate of CC).

| | | CC | | ACI-MP | RF-CP | S-CP | Fb-CP | |
|---|---|---|---|---|---|---|---|---|
| | | | | | | | ARA | IRA |
| Average cost | $\eta = 1000$ | 59.25 | $\alpha = 0.05$ | 17.970 | 15.794 | 17.321 | 15.356 | 7.189 |
| | $\eta = 500$ | 47.50 | $\alpha = 0.10$ | 17.263 | 14.378 | 16.17 | 14.228 | 6.798 |
| | $\eta = 100$ | 22.46 | $\alpha = 0.15$ | 16.096 | 11.922 | 14.83 | 12.354 | 6.191 |
| | $\eta = 50$ | 21.34 | $\alpha = 0.20$ | 15.310 | 10.032 | 13.217 | 10.22 | 5.398 |
| Average computation time | $\eta = 1000$ | 0.019 | $\alpha = 0.05$ | 0.022 | 0.487 | 0.022 | 0.027 | 0.038 |
| | $\eta = 500$ | 0.019 | $\alpha = 0.10$ | 0.026 | 0.494 | 0.020 | 0.021 | 0.039 |
| | $\eta = 100$ | 0.021 | $\alpha = 0.15$ | 0.021 | 0.545 | 0.021 | 0.020 | 0.037 |
| | $\eta = 50$ | 0.022 | $\alpha = 0.20$ | 0.022 | 0.500 | 0.020 | 0.019 | 0.036 |
| Collision avoidance rate | $\eta = 1000$ | 97.0% | $\alpha = 0.05$ | 98.6% | 98.7% | 98.8% | 98.2% | 96.3% |
| | $\eta = 500$ | 92.8% | $\alpha = 0.10$ | 93.3% | 96.9% | 93.5% | 94.6% | 94.1% |
| | $\eta = 100$ | 82.5% | $\alpha = 0.15$ | 91.5% | 92.4% | 92.0% | 90.2% | 91.9% |
| | $\eta = 50$ | 79.1% | $\alpha = 0.20$ | 87.9% | 90.0% | 88.2% | 86.7% | 88.2% |

and Fb-CP-IRA, the experimental results using the quadrotor model are fundamentally consistent with those derived from the experiments using the kinematic vehicle model. Specifically, compared with S-CP, Fb-CP-ARA benefits from the posterior probabilities calculation, leading to a moderate improvement in performance. Fb-CP-IRA, leveraging the combined use of posterior probabilities and a more flexible risk allocation, exhibits a significant enhancement in performance. Additionally, since CC and ACI-MP fail to fully utilize the information in the calibration dataset, they incur higher costs, which are 184% and 296% higher than those of Fb-CP-IRA, respectively. Particularly for CC, it directly controls the collision avoidance rate by adjusting the weight of the collision penalty term in the objective function, which results in a higher average cost. However, it should be noted that, in practice, CC and ACI-MP are better suited for scenarios where the test data exhibit distribution shift, rather than the setup considered in our work. For RF-CP, thanks to the proposed normalized nonconformity score, its average cost is comparable to that of Fb-CP-ARA, but it remains 85.8% higher than Fb-CP-IRA. However, the normalized nonconformity score introduces mixed-integer variables into the TO problem, significantly increasing the computation time. As shown in Table 3, the average computation time of RF-CP is more than an order of magnitude higher than that of Fb-CP-IRA.

### C.3 Simulation for dynamic bicycle model

We examine a vehicle with the following dynamic bicycle model Hakobyan and Yang [2021].

$$\dot{x} = v_x \cos\theta - v_y \sin\theta \tag{57}$$

$$\dot{y} = v_x \sin\theta + v_y \cos\theta \tag{58}$$

$$\dot{\theta} = r \tag{59}$$

$$\dot{v_y} = \frac{-2(C_f + C_r)}{m_V v_x} v_y - \left( \frac{2l_f C_f - 2l_r C_r}{m_V v_x} + v_x \right) r + \frac{2C_f}{m_V} \delta_f \tag{60}$$

$$\dot{r} = \frac{-2(l_f C_f + l_r C_r)}{I_z v_x} v_y - \frac{2l_f^2 C_f - 2l_r^2 C_r}{I_z v_x} r + \frac{2l_f C_f}{I_z} \delta_f \tag{61}$$

where $x, y$ are the vehicle's central of mass, $\theta, v_y$, and $r$ are lateral velocity, orientation, and yaw rate, respectively. Furthermore, $v_x$ is the constant longitudinal velocity, $m_V$ denotes the mass of the vehicle, $C_f$ and $C_r$ represent the cornering stiffness coefficients of the front and rear tires respectively, $L_f$ and $L_r$ denote the distances from the center of mass to the front and rear wheels, and $I_z$ corresponds to the moment of inertia around the $z$-axis. The input variable is the front wheel steering angle $\delta_f$. The system (57) is discretized using the sampling time $\Delta = 0.125$, and the total time is also set to $T = 20$. According to Hakobyan and Yang [2021], the parameters of the dynamic bicycle model used in this simulation are listed in Table 4.

Table 4: Dynamic bicycle model parameters.

| $m_V$ | $C_f$ | $C_r$ | $I_z$ | $L_f$ | $L_r$ | $v_x$ |
|---|---|---|---|---|---|---|
| $1700kg$ | $50kN/rad$ | $50kN/rad$ | $6000kg \cdot m^2$ | $1.2m$ | $1.3m$ | $5m/s$ |

The task is to steer the vehicle to its target point $p$ while avoiding $M = 2$ moving obstacles. Similar to the experiments in Appendix C.1, the target point constraint and obstacle avoidance constraints are incorporated into the optimization problem to ensure the vehicle reaches the target point while avoiding collisions with obstacles. We collect 13,000 joint obstacle trajectories and assign them as in Appendix C.1. The methods S-CP, Fb-CP-ARA, and Fb-CP-IRA are analyzed through 1,000 Monte Carlo simulations.

Table 5 shows the average cost, average computation time, and collision avoidance rate of 1,000 simulations using the dynamic bicycle model with different methods. The experimental results are generally consistent with those obtained from the experiments using the kinematic vehicle mode and the quadrotor model. The performance of Fb-CP shows a certain degree of improvement over S-CP based on posterior probability calculations. Based on posterior probability calculations, Fb-CP-ARA demonstrates a certain level of performance improvement compared to S-CP, while Fb-CP-IRA further attains significant performance by leveraging the combined use of posterior probabilities and a more flexible risk allocation. It should be noted that, due to the simulation of a relatively complex nonlinear model in this experiment, the average computation time inevitably increases. Furthermore, it may be observed that the reduction in average cost achieved by Fb-CP-IRA compared to S-CP decreases in this experiment (47.2% reduction) compared with the experiment using the kinematic vehicle model in Appendix C.1 (81.9% reduction). This is because, compared to relatively simple scenarios (2 obstacles, dynamic bicycle model experiment), more complex scenarios (3 obstacles, kinematic vehicle model experiment) better highlight the performance improvements enabled by the flexibility in risk allocation.

Table 5: Average cost, computation time, and collision avoidance rate using the dynamic bicycle model with different methods.

| | | S-CP | Fb-CP | |
| | | | ARA | IRA |
|---|---|---|---|---|
| | $\alpha = 0.05$ | 23.05 | 20.91 | 13.77 |
| | $\alpha = 0.10$ | 22.38 | 18.39 | 11.35 |
| Average cost | $\alpha = 0.15$ | 20.71 | 16.99 | 10.17 |
| | $\alpha = 0.20$ | 16.55 | 14.78 | 8.58 |
| | $\alpha = 0.05$ | 0.365 | 0.361 | 0.884 |
| | $\alpha = 0.10$ | 0.339 | 0.335 | 0.817 |
| Average computation time | $\alpha = 0.15$ | 0.494 | 0.506 | 1.292 |
| | $\alpha = 0.20$ | 0.309 | 0.407 | 1.003 |
| | $\alpha = 0.05$ | 96.8% | 96.5% | 97.0% |
| | $\alpha = 0.10$ | 94.8% | 94.0% | 94.3% |
| Collision avoidance rate | $\alpha = 0.15$ | 91.5% | 90.0% | 91.5% |
| | $\alpha = 0.20$ | 89.5% | 87.8% | 89.5% |

In summary, the three simulation experiments demonstrate the general applicability of the proposed method, achieving significant performance improvements across various system models while satisfying probabilistic collision avoidance requirements. In fact, the complexity of different system models only affects the average computation time. In addition, simulations demonstrate that Fb-CP-IRA achieves more significant performance improvements in relatively complex scenarios.

### C.4   Stanford Drone Dataset

We perform a comparative evaluation of different methods on the Stanford Drone Dataset. Specifically, the task is to steer the vehicle to its target point while avoiding moving obstacles (humans). Similar to the experiments in Appendix C.1, the target point constraint and obstacle avoidance constraints

are incorporated into the optimization problem to ensure the vehicle reaches the target point while avoiding collisions with obstacles. The average cost of different methods aresummarized in Table 6.

As shown in Table 6, compared with S-CP, Fb-CP-ARA benefits from the posterior probabilities calculation, resulting in an average cost reduction of at least 20.7%. Fb-CP-IRA reduces the cost by at least 46% compared to S-CP. Additionally, since CC and ACI-MP fail to fully utilize the information in the calibration dataset, they incur higher costs, which are 176% and 113% higher than those of Fb-CP-IRA, respectively. Note that RF-CP is not included as a baseline because its formulation introduces mixed-integer variables through nonconformity score definitions, which—when combined with the nonlinear vehicle dynamics—leads to highly complex trajectory optimization problems with prohibitively long computation times.

Table 6: Average cost of different methods on the Real-World Stanford Drone Dataset with different methods.

|  |  | CC |  | ACI-MP | S-CP | Fb-CP | |
|---|---|---|---|---|---|---|---|
|  |  |  |  |  |  | ARA | IRA |
| Average cost | $\eta = 1000$ | 95.51 | $\alpha = 0.05$ | 40.02 | 38.58 | 30.58 | 20.81 |
|  | $\eta = 500$ | 77.68 | $\alpha = 0.10$ | 35.94 | 34.75 | 27.52 | 18.52 |
|  | $\eta = 100$ | 41.27 | $\alpha = 0.15$ | 32.57 | 30.48 | 23.75 | 15.58 |
|  | $\eta = 50$ | 38.73 | $\alpha = 0.20$ | 29.89 | 28.59 | 21.25 | 14.02 |

## D   Experiment with different trajectory predictor

As noted in Subsection 3.1, employing the more advanced trajectory predictors can enhance control performance. Accordingly, we perform experiments on the kinematic vehicle model (54) to compare the average costs of S-CP and Fb-CP when using different predictors Social LSTM Alahi et al. [2016], Trajectron++ Salzmann et al. [2020], AgentFormer Yuan et al. [2021], with the results summarized in Table 7. As shown in Table 7, when switching to more advanced predictors Trajectron++ and AgentFormer, both Fb-CP and S-CP see improved performance, but the proposed method (Fb-CP) continues to outperform the S-CP under all tested predictors.

Table 7: Average Cost of Different Methods using Different Trajectory Predictors ($\alpha = 0.2$).

|  | S-CP | Fb-CP-ARA | Fb-CP-IRA |
|---|---|---|---|
| Social LSTM | 17.30 | 15.49 | 2.97 |
| Trajectron++ | 13.58 | 10.58 | 1.81 |
| AgentFormer | 12.83 | 9.71 | 1.72 |

## E   Experiment with small calibration size

As discussed in the Limitation (Appendix L), the proposed method requires a larger calibration dataset because the calibration dataset must be divided into two parts. To rigorously evaluate its performance under data-scarce conditions, we conduct experiments on the kinematic vehicle model (54), where both S-CP and Fb-CP are tested using the same small calibration dataset ($N = 400$). The corresponding average costs, computation time, and collision avoidance rate are reported in Table 8. As shown in Table 8, although all methods see a decrease in performance under limited data, Fb-CP still achieves substantially lower cost than S-CP and remains safety. This highlights the robustness and practical effectiveness of the proposed method, even when the calibration dataset is small.

## F   Experiment with higher-dimensional settings

In this appendix, we validate the performance of the proposed method in high-dimensional settings. Specifically, we conduct an additional case study on controlling the trajectory of a high-dimensional

Table 8: Average Cost, computation time, and collision avoidance rate of Different methods with calibration data size of $N = 400$.

| | S-CP | Fb-CP-ARA | Fb-CP-IRA |
|---|---|---|---|
| Average cost | 18.03 | 16.82 | 3.52 |
| Average computation time | 0.078 | 0.081 | 0.122 |
| Collision avoidance rate | 88.9% | 89.5% | 91.7% |

double integrator model, where shrinking-horizon trajectory optimization is performed in 5- and 10-dimensional spaces. The average costs and collision probabilities of different methods under various dimensions are summarized in Table 9. As shown in Table 9, the proposed Fb-CP framework consistently outperforms S-CP while maintaining safety, even in these higher-dimensional scenarios. These results validate the applicability and effectiveness of the proposed approach in high-dimensional problems.

Table 9: Average cost and collision avoidance rate of different methods ($\alpha = 0.2$) with varying dimensions.

| | Dimension | S-CP | Fb-CP-ARA | Fb-CP-IRA |
|---|---|---|---|---|
| Average cost | 5 | 44.52 | 39.67 | 30.25 |
| | 10 | 72.66 | 65.33 | 60.18 |
| Collision avoidance rate | 5 | 92.5% | 92.1% | 93.5% |
| | 10 | 91.2% | 93.5% | 91.4% |

# G   Extension and experiments on distribution shift

The individual chance constraint reformulation in Lemma 4.1 and the posterior probability calculation in Lemma 4.2 rely on Assumption 3.1 and the i.i.d. property of $\omega$, which imply that real joint obstacle trajectory and those in the training and calibration datasets follow the same distribution. To enhance the generalizability of the proposed method across different scenarios, we extend our method to address the case where a shift exists between the initial state distributions of calibration data and test data, and the error $\omega_t$ represents state-dependent noise rather than being i.i.d.

Specifically, there exists a shift between the initial obstacle state distribution $\mathcal{D}_{test}$ of the test data and the initial obstacle state distribution $\mathcal{D}_{cal}$ of the calibration data. Moreover, the error $\omega_t$ constitutes state-dependent noise, i.e., its distribution $\mathcal{D}_{\omega_t}$ is conditioned on the current state $Y_t$. Clearly, the difference between $\mathcal{D}_{test}$ and $\mathcal{D}_{cal}$, along with the state-dependent nature of $\omega_t$, implies that the test trajectories $Y_t$ and the calibration trajectories $Y_t^{(i)}$ are not exchangeable. Fortunately, the distribution shift between $Y_t$ and $Y_t^{(i)}$ can be characterized as a covariate shift, as defined in Tibshirani et al. [2019]. This allows us to adapt the approach proposed in Tibshirani et al. [2019] to extend our method accordingly. The extended method is described in detail below.

Intuitively, we reweight the calibration data by computing the likelihood ratio between the test and calibration distributions. The reweighted data are then used to compute the prediction regions and the posterior probabilities, thereby enabling robustness to covariate shift. Specifically, at each time $t$, we begin by applying an uncertainty propagation technique to derive the distributions $\widetilde{\mathcal{D}}_{Y_t}$ and $\mathcal{D}_{Y_t}$ of the test obstacle state $Y_t$ and calibration obstacle state $Y_t^{(i)}$, respectively. As a result, given a data $Y$, we can calculate the likelihood ratio as follows.

$$v(Y) = \mathrm{d}\mathbb{P}_{\widetilde{\mathcal{D}}_{Y_t}}(Y)/\mathrm{d}\mathbb{P}_{\mathcal{D}_{Y_t}}(Y) \tag{62}$$

In the forward phase of confidence region computation, the weights for the test data and the calibration data in $D_{cal}^1$ are computed using the likelihood ratios (62) as follows.

$$p_1(Y_t) = \frac{v(Y_t)}{\sum_{j=1}^{K} v(Y_t^{(j)}) + v(Y_t)}, \quad p_1(Y_t^{(i)}) = \frac{v(Y_t^{(i)})}{\sum_{j=1}^{K} v(Y_t^{(j)}) + v(Y_t)}, \quad \forall i = 1, ..., K \tag{63}$$

Following the approach in Tibshirani et al. [2019], we replace the $(1 - \alpha)$-quantile in equation 6b with the following weighted form.

$$C_{\tau|t}^{1-\alpha_\tau} = Quantile_{1-\alpha_\tau} \left( \sum_{i=1}^{K} p_1(Y_t^i)\delta_{R_{\tau|t}^{(i)}} + p_1(Y_t)\delta_\infty \right) \tag{64}$$

In the process of backward posterior probability computation, we calculate the weights for the test data and the calibration data in $D_{cal}^2$ as follows.

$$p_2(Y_t) = \frac{v(Y_t)}{\sum_{j=K}^{K+L} v(Y_t^{(j)}) + v(Y_t)}, \quad p_2(Y_t^{K+i}) = \frac{v(Y_t^{(K+i)})}{\sum_{j=K}^{K+L} v(Y_t^{(j)}) + v(Y_t)}, \quad \forall i = 1, ..., L \tag{65}$$

For the posterior probability computation, we replace equation (8) in Lemma 4.2 with the following weighted form.

$$\mathbb{P}\{c(x_\tau^*, Y_\tau) < 0\} \leq \beta_\tau = p_2(Y_t) + \sum_{i=1}^{L} p_2\left(Y_t^{K+i}\right) \mathbb{I}\left(S_\tau^{(K+i)} < 0\right) \tag{66}$$

It is evident that when $p_2(Y_t) = p_2(Y_t^{K+i}) = 1/(L+1)$, equation (66) degenerates to equation (8). By applying the weighting schemes in equations (65) and (66) to the test and calibration data, the extended method is capable of handling the covariate shift between them. We refer to this extended approach as Weighted Fb-CP.

To demonstrate the Weighted Fb-CP exhibits robustness to the covariate shift, we design experiments to compare the effects of covariate shift between test trajectories and calibration trajectories on the performance and safety of the proposed method. Apart from the method of generating obstacle trajectories, the experimental setup is identical to that of the kinematic vehicle model experiment in Appendix C.1. To obtain obstacle trajectories with different distributions, the obstacles are modeled using the following double integrator model.

$$\begin{bmatrix} p_{x,t+1} \\ p_{y,t+1} \\ v_{x,t+1} \\ v_{y,t+1} \end{bmatrix} = \begin{bmatrix} p_{x,t} + \Delta v_{x,t} + \frac{\Delta^2}{2}a_{x,t} \\ p_{y,t} + \Delta v_{y,t} + \frac{\Delta^2}{2}a_{y,t} \\ v_{x,t} + \Delta a_{x,t} \\ v_{y,t} + \Delta a_{y,t} \end{bmatrix} + \omega_t e \tag{67}$$

where $(p_x, p_y, v_x, v_y)$ is the state of an obstacle, consisting of its center of mass and velocity vector. The control input $u = (a_x, a_y)$ is the acceleration vector. Similarly, the sampling time $\Delta$ is selected as 0.125. The obstacle trajectories from a given start point to the target point are obtained by solving an optimization problem. $\omega_t$ is sampled from a zero-mean Gaussian distribution, with its variance determined by the current state $(p_x, p_y, v_x, v_y)$. The initial states of the trajectories in the test dataset are sampled from a Gaussian distribution with mean $(-5, 0, 0, 0)^T$, while the initial states of the trajectories in the calibration dataset are sampled from a Gaussian distribution with mean $(0, -5, 0, 0)^T$. As a result, there exists a distribution shift between the trajectories in the test dataset and the calibration dataset. We conducted 1,000 Monte Carlo experiments in this scenario to compare the Weighted Fb-CP, Fb-CP, S-CP Lindemann et al. [2023], and ACI-MP Dixit et al. [2023].

Table 10: Average cost, computation time, and collision avoidance rate using the kinematic vehicle model with different methods ($\alpha = 0.2$).

|  | ACI-MP | S-CP | Fb-CP | | Weighted Fb-CP | |
| --- | --- | --- | --- | --- | --- | --- |
|  |  |  | ARA | IRA | ARA | IRA |
| Average cost | 19.27 | 17.56 | 14.35 | 11.37 | 16.27 | 13.72 |
| Collision avoidance rate | 84.8% | 78.7% | 78.6% | 76.8% | 84.6% | 83.4% |

Table 10 shows the average cost and collision avoidance rate of 1,000 simulations using the kinematic vehicle model with different methods. It can be observed that due to the covariate shift, both the S-CP and the proposed Fb-CP methods exceed the specified risk tolerance ($\alpha = 0.2$). Owing to the reweighting of the test and calibration data, the Weighted Fb-CP method is able to maintain the

collision probability within the prescribed risk tolerance, accompanied by a justifiable and expected increase in average cost. For ACI-MP, since it does not rely on the calibration set, it naturally does not exceed the risk tolerance. However, precisely because it cannot leverage the information contained in the calibration dataset, its average cost is 40.5% higher than that of Weighted Fb-CP-IRA. In summary, we empirically demonstrate that the Weighted Fb-CP is capable of maintaining the risk probability below the specified tolerance under covariate shift, while also achieving a favorable trade-off in terms of average cost.

## H  Details about the prediction regions

In this Section, we investigate the impact of using prior versus posterior probabilities on the prediction regions. To this end, we collect the prediction region radius for time $t$, denoted as $C_{20|t}$, using the vehicle model with different methods across 1,000 simulations, as illustrated in Figure 3. It can be observed that $C_{20|t}$ decreases as $t$ increases, which is reasonable since the error of the trajectory predictor diminishes as $t$ approaches $\tau$. Note that since S-CP only uses prior probabilities to compute $C_{20|t}$ throughout the entire planning process, which depends solely on $D_{cal}$, $C_{20|t}$ remains constant for a fixed $t$. By contrast, for Fb-CP-ARA, $C_{20|t}$ also depends on the actual obstacle positions and past decisions due to the use of the posterior probabilities, which leads to the variability of $C_{20|t}$ across 1,000 simulations. The distribution of $C_{20|9}$ is shown in the right panel of Figure 3. It can be seen that $C_{20|t}$ computed by Fb-CP-ARA is typically smaller than that computed by S-CP. As $t$ increases, more posterior probabilities can be used, leading to a growing gap between the $C_{20|t}$ calculated by the two methods, which corroborates Corollary A.2.

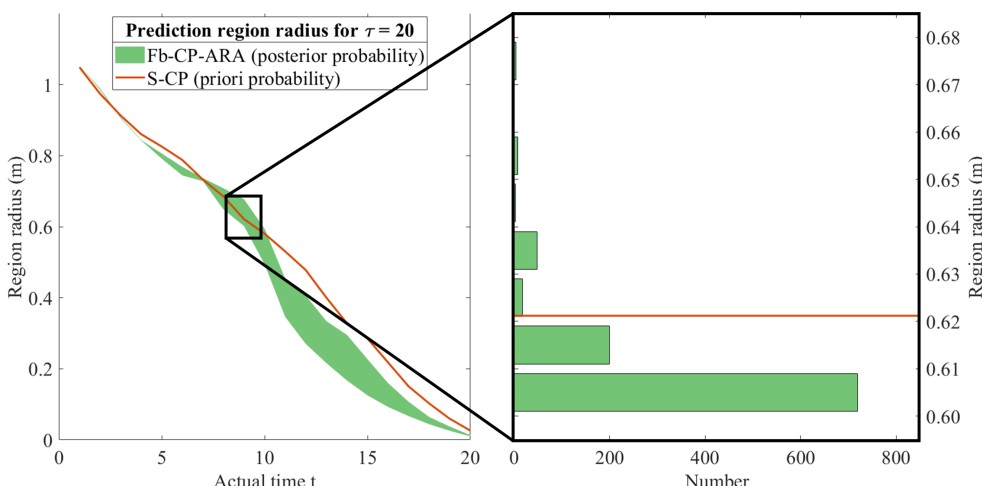

Figure 3: Left: prediction region radius for $\tau = 20$ at each time $t$ ($C_{20|t}$) using the vehicle model with different methods across 1,000 simulations. Right: distributions of $C_{20|9}$.

Furthermore, Table 11 shows the prediction region radius for different time $t$ and $\tau$ ($C_{\tau|t}$). At the initial state ($t = 0$), no realized state is available for calculating posterior probabilities. As a result, $C_{\tau|0}$ for all $\tau$ obtained by S-CP and Fb-CP-ARA are essentially identical, with minor differences arising from the fact that S-CP utilizes the calibration $\mathcal{D}_{cal}$, whereas Fb-CP-ARA only employs $\mathcal{D}_{cal}^1$. As the system operates, an increasing number of realized states $x_t^*$ are available for the calculation of posterior probabilities, enabling Fb-CP to yield a relatively narrower prediction region, corresponding to a smaller $C_{\tau|t}$. As shown in Table 11, the average ratio of the predicted region radius obtained by Fb-CP-ARA to those by S-CP generally exhibits a decreasing trend as time $t$ increases. Moreover, when $t > 10$, based on sufficient posterior probabilities, the prediction region radius obtained by Fb-CP-ARA is reduced by more than $50\%$ compared to that of S-CP.

Table 11: Prediction region radius for different $\tau$ and $t$ ($C_{\tau|t}$) using the kinematic vehicle model with different methods (S-CP and Fb-CP-ARA) across 1,000 simulations ($\alpha = 0.2$).

| | | $\tau = 3$ | $\tau = 6$ | $\tau = 9$ | $\tau = 12$ | $\tau = 15$ | $\tau = 18$ | Ratio |
|---|---|---|---|---|---|---|---|---|
| $t = 0$ | Fb-CP-ARA | 0.125 | 0.258 | 0.374 | 0.519 | 0.698 | 0.913 | 1.005 |
| | S-CP | 0.129 | 0.247 | 0.374 | 0.520 | 0.691 | 0.902 | |
| $t = 3$ | Fb-CP-ARA | ╲ | 0.102 | 0.216 | 0.351 | 0.527 | 0.705 | 0.793 |
| | S-CP | | 0.129 | 0.263 | 0.409 | 0.598 | 0.738 | |
| $t = 6$ | Fb-CP-ARA | ╲ | ╲ | 0.141 | 0.289 | 0.442 | 0.604 | 0.956 |
| | S-CP | | | 0.152 | 0.304 | 0.451 | 0.625 | |
| $t = 9$ | Fb-CP-ARA | ╲ | ╲ | ╲ | 0.134 | 0.268 | 0.414 | 0.889 |
| | S-CP | | | | 0.149 | 0.305 | 0.466 | |
| $t = 12$ | Fb-CP-ARA | ╲ | ╲ | ╲ | ╲ | 0.059 | 0.139 | 0.453 |
| | S-CP | | | | | 0.138 | 0.291 | |
| $t = 15$ | Fb-CP-ARA | ╲ | ╲ | ╲ | ╲ | ╲ | 0.054 | 0.470 |
| | S-CP | | | | | | 0.115 | |

# I Computation time of IRA and the hybrid method of ARA and IRA

Table 12 shows the average computation time at each time $t$ using the kinematic vehicle model with Fb-CP-IRA ($\alpha = 0.2$). It can be observed that, in practice, the majority of the additional computational burden introduced by IRA arises at the initial time $t = 0$, where it is used to obtain the initial risk allocation and initial trajectory. At subsequent time steps, by using the optimal solution from the previous time step (or iteration) as the initial value for the next time step (or iteration), the IRA algorithm can converge quickly. It is important to note that the TO problem at the initial time step can be solved offline, while the average computation time for TO at subsequent time steps is much smaller than the sampling time ($0.125s$). Therefore, Fb-CP-IRA is well-suited for real-time TO.

Table 12: Average Computation Time (ACT) at each time $t$ using the kinematic vehicle model with Fb-CP-IRA ($\alpha = 0.2$).

| $t$ | 0 | 1 | 2 | 3 | 4 | 5 | 6 | 7 | 8 | 9 |
|---|---|---|---|---|---|---|---|---|---|---|
| ACT | 2.302 | 0.035 | 0.033 | 0.032 | 0.028 | 0.026 | 0.024 | 0.022 | 0.021 | 0.022 |

| $t$ | 10 | 11 | 12 | 13 | 14 | 15 | 16 | 17 | 18 | 19 |
|---|---|---|---|---|---|---|---|---|---|---|
| ACT | 0.015 | 0.011 | 0.010 | 0.009 | 0.008 | 0.007 | 0.006 | 0.005 | 0.005 | 0.002 |

Furthermore, we explore a hybrid method of ARA and IRA to achieve a trade-off between average cost and average computation time. Specifically, we define a switching time $t_s$, such that when $t < t_s$, the IRA method is used, and when $t \geq t_s$, the ARA method is applied. Table 13 shows the average cost and computation time using the kinematic vehicle model with different switching time $t_s$ ($\alpha = 0.2$). It can be observed that using IRA only at the initial time step results in a 66.56% reduction in average cost. This is because the trajectory obtained at the initial time step determines the overall path of the entire trajectory. As $t_s$ increases, the average cost naturally decreases. When $t_s$ reaches 11, further increases in $t_s$ no longer lead to significant reduction in the average cost. This is because, in our scenario, the interaction between obstacles and the vehicle is most intensive at the middle of the mission time. After $t > 11$, the obstacles and the vehicle have moved apart, significantly reducing the collision risk, which results in ARA and IRA optimizing nearly identical trajectory. Although the aforementioned hybrid method balances the trade-off between average cost and computation time, as previously mentioned, executing IRA at the initial time step is the primary source of both cost reduction and increased computation time.

Table 13: Average cost and computation time using the kinematic vehicle model with different switching time $t_s$ ($\alpha = 0.2$).

| $t_s$ | 0 (ARA) | 1 | 6 | 11 | 16 | 20 (IRA) |
|---|---|---|---|---|---|---|
| Average cost | 15.13 | 5.06 | 3.74 | 2.90 | 2.89 | 2.89 |
| Average computation time | 0.078 | 0.118 | 0.121 | 0.129 | 0.129 | 0.131 |

## J  Sensitivity analysis of the calibration set division

Table 14 shows the average cost, average computation time and collision avoidance rate using the kinematic vehicle model and Fb-CP-ARA ($\alpha = 0.2$) with 10 different random calibration set divisions. Specifically, for each experiment, we randomly divide the calibration dataset $D_{cal}$ into $D_{cal}^1$ and $D_{cal}^2$. It can be observed that, for the ten random experiments, the standard deviation of the average cost is 0.643, and the coefficient of variation is 4.2%, indicating relatively low volatility. For the collision avoidance rate, its volatility is only 2.3%, and all values do not exceed the given tolerance (80%). Therefore, the proposed method is insensitive to the calibration set division.

Table 14: Average Cost (AC), Average Computation Time (ACT) and Collision Avoidance Rate (CAR) using the kinematic vehicle model and Fb-CP-ARA ($\alpha = 0.2$) with 10 different random calibration set divisions.

| Index | 1 | 2 | 3 | 4 | 5 | 6 | 7 | 8 | 9 | 10 |
|---|---|---|---|---|---|---|---|---|---|---|
| AC | 15.13 | 14.78 | 15.11 | 16.03 | 15.96 | 14.37 | 14.60 | 15.73 | 15.64 | 16.19 |
| ACT | 0.078 | 0.078 | 0.079 | 0.077 | 0.072 | 0.080 | 0.065 | 0.074 | 0.086 | 0.084 |
| CAR | 89.1% | 89.2% | 90.5% | 88.2% | 88.6% | 88.4% | 90.5% | 89.2% | 89.3% | 89.9% |

## K  An extension using the normalized nonconformity score

Stamouli et al. [2024] proposed a normalized nonconformity score, which can improve performance compared to S-CP [Lindemann et al., 2023]. In this section, we incorporate this normalized nonconformity score into Fb-CP to further enhance its performance as follows. First, we still follow the content outlined prior to Section 4.1. Instead of using the original nonconformity score as in Section 4.1, we can redefine the nonconformity score at time $\tau$ as follows to replace (5).

$$
\begin{aligned}
R_\tau &= \max_{t=0,\ldots,\tau-1} \left\{ \frac{\|Y_\tau - \hat{Y}_{\tau|t}\|}{\sigma_{\tau|t}} \right\} \\
R_\tau^{(i)} &= \max_{t=0,\ldots,\tau-1} \left\{ \frac{\|Y_\tau^{(i)} - \hat{Y}_{\tau|t}^{(i)}\|}{\sigma_{\tau|t}} \right\} \quad \forall i = 1,\ldots,K
\end{aligned}
\tag{68}
$$

where

$$
\sigma_{\tau|t} = \max_{j \in \mathcal{I}_{train}} \|Y_\tau^{(j)} - \hat{Y}_{\tau|t}^{(j)}\|, \quad \forall t, \ \tau > t
\tag{69}
$$

where $\mathcal{I}_{train} = \{j : Y^{(j)} \in D_{train}\}$ denotes the set of indices of the data in the training set $D_{train}$. We note that, compared to the nonconformity score in Stamouli et al. [2024], we separate the nonconformity score at each time $\tau$, which facilitates the reallocation of the risk at each time step. Similarly, given an allocated risk $\alpha_\tau$ for future time $\tau$, the random variables $R_\tau, R_\tau^{(1)}, \ldots, R_\tau^{(K)}$ are exchangeable and the prediction region with coverage guarantee is derived as follows.

$$
\mathbb{P}\left\{ \max_{t=0,\ldots,\tau-1} \left\{ \frac{\|Y_\tau - \hat{Y}_{\tau|t}\|}{\sigma_{\tau|t}} \right\} \le C_\tau^{1-\alpha_\tau} \right\} \ge 1 - \alpha_\tau
\tag{70a}
$$

$$
C_\tau^{1-\alpha_\tau} = Quantile_{1-\alpha_\tau}(R_\tau^{(1)}, \ldots, R_\tau^{(K)}, \infty)
\tag{70b}
$$

Based on the $(1 - \alpha_\tau)$-coverage prediction region defined in (70a), the individual chance constraint $\mathbb{P}\{c(x_\tau, Y_\tau) \ge 0\} \ge 1 - \alpha_\tau$ can be reformulated as the following lemma.

**Lemma K.1** *(chance constraint) If Assumption 3.1 holds, the constraint function $c$ is $L$-Lipschitz continuous and $\max_{0 \le s \le t}\{c(x_\tau, \hat{Y}_{\tau|t}) - LC_{\tau|t}^{1-\alpha_\tau}\} \ge 0$ is satisfied where $C_{\tau|t}^{1-\alpha_\tau} = \sigma_{\tau|t} C_\tau^{1-\alpha_\tau}$, then the individual chance constraint $\mathbb{P}\{c(x_\tau, Y_\tau) \ge 0\} \ge 1 - \alpha_\tau$ is satisfied.*

*Proof:* According to the $(1 - \alpha_\tau)$-coverage prediction region defined in 70a, we can obtain that

$$\mathbb{P}\left\{ \bigcap_{t=0}^{\tau-1} \left\{ \frac{\|Y_\tau - \hat{Y}_{\tau|t}\|}{\sigma_{\tau|t}} \right\} \le C_\tau^{1-\alpha_\tau} \right\} \ge 1 - \alpha_\tau \tag{71}$$

According to that fact $t \le \tau - 1$ and the definition, we have the following inequality.

$$\mathbb{P}\left\{ \bigcap_{t=0}^{t} \left\{ \|Y_\tau - \hat{Y}_{\tau|t}\| - C_{\tau|t}^{1-\alpha_\tau} \right\} \le 0 \right\} \ge \mathbb{P}\left\{ \bigcap_{t=0}^{\tau-1} \left\{ \|Y_\tau - \hat{Y}_{\tau|t}\| - C_{\tau|t}^{1-\alpha_\tau} \right\} \le 0 \right\} \ge 1 - \alpha_\tau \tag{72}$$

Based on (72), we can further obtain the following inequality.

$$\mathbb{P}\left\{ C_{\tau|s}^{1-\alpha_\tau} - \|Y_\tau - \hat{Y}_{\tau|s}\| \ge 0 \right\} \ge 1 - \alpha_\tau, \quad \forall s = 0, ..., t \tag{73}$$

Note that the function $c$ is $L$-Lipschitz continuous, the following inequality is obtained.

$$\|c(x_\tau, Y_\tau) - c(x_\tau, \hat{Y}_{\tau|t})\| \le L\|Y_\tau - \hat{Y}_{\tau|t}\| \implies c(x_\tau, Y_\tau) \ge c(x_\tau, \hat{Y}_{\tau|t}) - L\|Y_\tau - \hat{Y}_{\tau|t}\| \tag{74}$$

If the constraint $\max_{0 \le s \le t}\{c(x_\tau, \hat{Y}_{\tau|t}) - LC_{\tau|t}^{1-\alpha_\tau}\} \ge 0$ is satisfied, we have the following inequality.

$$\exists s = 0, ..., t \quad c(x_\tau, Y_\tau) \ge L(C_{\tau|s}^{1-\alpha_\tau} - \|Y_\tau - \hat{Y}_{\tau|s}\|) \tag{75}$$

By combining (73) and (75), we ultimately obtain $\mathbb{P}\{c(x_\tau, Y_\tau) \ge 0\} \ge 1 - \alpha_\tau$.

Finally, it is sufficient to replace constraint (11e) in the TO problem (11) with $\max_{0 \le s \le t}\{c(x_\tau, \hat{Y}_{\tau|t}) - LC_{\tau|t}^{1-\alpha_\tau}\} \ge 0$. For the posterior probability calculation and risk allocation method, since we have separated the nonconformity score at each time step $\tau$, our proposed framework remains fully applicable. It is important to note that, as described in Section C, the constraint $\max_{0 \le s \le t}\{c(x_\tau, \hat{Y}_{\tau|t}) - LC_{\tau|t}^{1-\alpha_\tau}\} \ge 0$ introduces mixed-integer variables into the TO problem, which significantly increases the solution time, especially when using the IRA method and dealing with nonlinear systems.

## L    Limitations

The main limitation lies in the reliance of the proposed method on the size of the calibration dataset. As previously mentioned, to ensure coverage guarantees within the closed-loop framework, the calibration dataset needs to be split into two parts: one for forward computation of prediction regions and the other for backward computation of posterior probabilities. This requirement results in the proposed method needing a larger calibration dataset compared to standard CP methods. However, extensive data can be sourced from advanced high-fidelity simulators or robotic applications like autonomous vehicles, where datasets are increasingly accessible. Thus we believe that the reliance on data quantity will not present a substantial challenge.

## M    Broader Impacts

This work proposes a novel Fb-CP framework for trajectory optimization under uncertainty, with provable safety guarantees and adaptive risk control. The method has potential positive societal impacts on safety-critical applications such as autonomous vehicles, robotics, and disaster response, by improving the reliability and efficiency of decision-making under uncertainty. It also contributes to the development of trustworthy AI through its theoretical guarantees and feedback-based adaptability.

However, potential negative societal impacts include misuse in high-risk or adversarial settings (e.g., autonomous weapons), privacy concerns from trajectory data collection, and overconfident decisions if the system fails under distribution shift. To mitigate these risks, future deployments should incorporate rigorous validation, privacy safeguards, and oversight mechanisms to ensure safe and ethical use of the proposed method.

