# OpenReview forum: "Conformal Prediction in The Loop: A Feedback-Based Uncertainty Model for Trajectory Optimization"
_NeurIPS.cc/2025/Conference — NeurIPS 2025 poster_

### Official Review · Reviewer_4FW9 · 2025-06-23

**Clarity:** 2
**Significance:** 3
**Originality:** 2
**Rating:** 4
**Confidence:** 4

**Summary:**

The paper addresses the problem of complementing time-series prediction with reliable prediction intervals using efficient conformal prediction sets. To improve the efficiency of these sets, the authors propose re-estimating the miscoverage rate at past time steps in order to reallocate the budget for future steps. Across a range of experiments, the proposed method outperforms baseline approaches that fix the miscoverage requirement beforehand and do not adapt it over time.

**Questions:**

- CP sets are known to have two-sided guarantees and to be asymptotically optimal. That is, one can show that the coverage probability is also lower bounded, and that the bound approaches the target level $\alpha$ as the sample size $n$ increases. Given this, it seems that ‘recalibrating’ the thresholds is not necessary for large $n$, as these sets are not conservative in that regime. Where does the conservativeness of CP arise in the scenario considered in the paper? Why would one need to use the proposed method for large $n$?

-I believe Remark 4.4 should be toned down. Corollary 4.3 shows that the expected value of the adjusted reliability level is lower than a quantity that depends on the original value. However, it is not straightforward to conclude that $\beta < \alpha$, especially for small $N$ and $L$. Claiming that the proposed approach performs no worse than the non-adjusted one seems to be a stretch.

Some minor mistakes:
- line 200 "an feasible..."
- line 230 "c(.)"
- In eq.9  should not $x_\tau^*$ depend on (i) since it is the policy for the specific trajectory?

**Ethical Concerns:**

["NO or VERY MINOR ethics concerns only"]

**Final Justification:**

The authors have addressed my concerns. I have updated my score.

**Limitations:**

- The main limitation of this work is the i.i.d. residual error assumption, which is quite restrictive in time-series forecasting. This is especially true given that the paper deals with CP, whose main strength lies in being a distribution-free framework.

**Paper Formatting Concerns:**

No issues

**Quality:**

3

**Strengths And Weaknesses:**

The idea of "stacking" CP predictions to reliably estimate the conservativeness of the policy is interesting and novel. In practice, this is shown to greatly improve the performance of time-series CP without compromising coverage.

My main concerns about the paper are as follows:
- The assumption of i.i.d. prediction errors is quite restrictive, as one would expect the errors to depend on past quantities. The main text mentions that this assumption can be removed, and Appendix D provides further discussion. However, based on the material in Appendix D, it is unclear how one would extend Lemma 4.2 and Corollary 4.3 once this assumption is dropped.
- The guarantees of the algorithm are not clearly stated. The original "uncorrected" coefficients are guaranteed to produce policies that satisfy the constraint requirements. Do the same guarantees hold for the adjusted coefficients?”

---

> ### Author Rebuttal · Authors · 2025-07-29
>
> Thank you very much for your valuable comments, which have helped us to improve the quality of this manuscript.
>
> >**Weakness 1.**
>
> We thank the reviewer for this important and insightful comment. Following your suggestion, we carefully revisited the theoretical assumptions and proofs throughout the paper, and found that **the i.i.d. assumption on prediction errors $\\omega\_{t}$ is indeed unnecessarily strong and can be significantly relaxed, as we now explain.**
>
> Specifically, it is not necessary to assume that the prediction errors at different time steps within the same trajectory are i.i.d.. Instead, we only require that the prediction errors at the same time step $\tau$ across different trajectories (i.e., test and calibration) are i.i.d.. This weaker condition holds under the following setup: all trajectories are generated by sampling their initial state $Y\_{0}^{(i)}$ from the same distribution $\\mathcal{D}$ and then evolving using a common dynamics $Y\_{t+1}=g(Y\_{t})+\\omega\_{t}$. Importantly, this places no restriction on the temporal dependence within the prediction errors $\\omega\_{1} ,…, \\omega\_{T−1}$ along a single trajectory. Therefore, errors that depend on past quantities are fully allowed within each trajectory, addressing the reviewer's concern.
>
> To further clarify, the reason this weaker assumption is sufficient lies in the requirement of exchangeability when applying conformal prediction. In the proof of Lemma 4.2, what we need is for the nonconformity scores
> $$S\_{\\tau}  = c(x\_{\\tau}^{\*},\\hat{Y}\_{\\tau|\\tau-1} + \\omega_{\\tau}), \\quad S\_{\\tau}^{(i)} = c(x\_{\\tau}^{\*},\\hat{Y}\_{\\tau|\\tau-1} + \\omega\_{\\tau}^{(i)}) \\quad \\forall i=K+1,...,K+L$$
> to be exchangeable. This is satisfied as long as the prediction errors at time $\\tau$ across trajectories are i.i.d.—**no assumption is needed on how $\\omega$ evolves over time within the same trajectory**.
>
> As for Appendix D, we acknowledge that it presents only a practical approach to handling scenarios where test and calibration trajectories may come from different dynamics models. This is meant to be an empirical workaround when the required exchangeability is violated. However, we agree that this is not a substitute for a rigorous theoretical extension, and have clarified this point more explicitly in the revised version.
>
> We also emphasize that the obstacle trajectories used in our experiments were either generated using the trajectory simulator TrajNet++ [1] or taken from the real-world Stanford Drone Dataset [2], both of which are temporally correlated time-series data. Our framework achieved strong performance while maintaining safety, further validating its practical applicability under realistic conditions.
>
> We sincerely thank the reviewer again for raising this question. It helped us **identify and remove an unnecessarily strong assumption**, thus improving both the clarity and generality of the theoretical framework.
>
> >**Weakness 2.**
>
> We thank the reviewer for pointing out this important concern. We confirm that **our framework always guarantees that the entire planned trajectory satisfies the overall collision risk tolerance**, even when using adjusted risk allocations based on posterior probability. Below, we explain the guarantee in more detail.
>
> At each time step $t$, the realized state up to that point, $x\_{1}^{∗},…,x\_{t}^{∗}$, is already available. Using Lemma 4.1, we can compute the posterior probabilities $\\beta\_{1},…,\\beta\_{t}$ for these realized positions. When optimizing the remainder of the trajectory for future steps $\\tau=t+1,…,T$, we allow risk reallocation over these future time steps by adjusting $\\alpha\_{t+1},…,\\alpha\_{T}$. It is important to note that the total risk allocation always satisfies $\\sum_{\\tau=1}^{t} \\beta\_{\\tau} + \\sum_{\\tau=t+1}^{T} \\alpha\_{\\tau} \\leq \alpha$, where $\\alpha$ is the overall risk tolerance. This ensures that the cumulative collision risk over the full trajectory remains within $\\alpha$, regardless of how the remaining risk is redistributed. For $\\tau>t$, we again apply Lemma 4.1 to convert the allocated risk $\\alpha\_{\\tau}$ into a deterministic constraint. These constraints are enforced during optimization, ensuring that each planned state $x\_{\\tau}^{∗}$ has a collision probability no greater than $\\alpha\_{\\tau}$.
>
> Therefore, even with dynamically adjusted coefficients, the entire trajectory remains safe under the risk threshold. We acknowledge that this guarantee was not made sufficiently explicit in the original paper. In the revised manuscript, we have now clearly stated this theoretical guarantee to improve clarity.
>
> >**Question 1.**
>
> We thank the reviewer for these thoughtful questions. We believe there may be a misunderstanding regarding our Fb-CP framework. **The primary function of our posterior recalibration is not to correct for potential over-conservativeness of CP sets due to small $n$, but rather to exploit information that becomes available during execution—specifically, the realized state $x^{*}$—to tighten the constraint violation probability estimate.**
>
> To clarify this with an example, we consider a specific chance constraint $\\mathbb{P} \\{ \\Vert x\_{\\tau}-Y\_{\\tau}\Vert -r\\}\\geq 1-\\alpha_{\\tau}$ . During the forward conformal step, Lemma 4.1 converts this chance constraint into a deterministic constraint of the form $\\Vert x\_{\\tau}-\\hat{Y}\_{\\tau|t}\Vert -r-C\_{\\tau|t}^{1-\\alpha_{\\tau}} \\geq 0$, where $C\_{\\tau|t}^{1-\\alpha_{\\tau}}$ is the conformal correction derived from the calibration set $D\_{cal}^{1}$. This constraint is then enforced in the trajectory optimizer, resulting in a realized state $x\_{\\tau}^{∗}$ that satisfies the above chance constraint. According to standard CP theory, this guarantee is essentially achieved by ensuring that at least a $1-\\alpha\_{\\tau}$ fraction of samples in $D\_{cal}^{1}$ fall within the region $\\{p : \Vert p-\\hat{Y}\_{\\tau|t} \Vert - r - C\_{\\tau|t}^{1-\\alpha\_{\\tau}} < 0 \\}$. However, it is important to note that not all samples outside this region would actually result in a collision with the realized state $x\_{t}^{∗}$. In other words, the original constraint enforces safety without knowledge of the actual realization $x\_{t}^{∗}$, leading to inherent conservativeness.
>
> Our feedback mechanism (Lemma 4.2) directly addresses this: after observing $x\_{t}^{∗}$, we evaluate the true violation rate on $D\_{cal}^{2}$, i.e., the fraction of samples that truly violate $\\Vert x\_{t}^{*} - Y_{t}^{(i)}\\Vert-r \geq 0$, which yields a more precise posterior estimate of the actual risk.
>
> Therefore, the conservativeness of the prior CP constraint arises not from limited sample size $n$, but rather from its inability to incorporate the realized decision $x\_{\tau}^{*}$ at planning time. This structural conservativeness remains even when $n$ is large, and our method addresses it directly. The benefit of posterior adjustment is thus present regardless of the dataset size, as it leverages new information unavailable in the original conformal calibration step. We thank the reviewer again for prompting this clarification.
>
> >**Question 2.**
>
> We thank the reviewer for the constructive feedback. We fully agree that drawing a rigorous performance guarantee solely from Corollary 4.3 is too strong, particularly for small calibration sizes $N$ and $L$. Indeed, the improved performance of our posterior-adjusted method primarily stems from leveraging the realized state, which is not directly captured in Corollary 4.3. Therefore, following the reviewer’s suggestion, we have:
> - Removed the claim that the proposed approach is theoretically no worse than the unadjusted baseline;
> - Retained only the empirical observation that our method tends to improve performance in practice;
> - Moved Remark 4.4 to the appendix, where it serves as a supplemental empirical insight.
>
> We thank the reviewer again for helping us clarify the limitations of this theoretical interpretation and improve the precision of our manuscript.
>
> >**Minor mistakes.**
>
> Thank you for pointing out the typos and raising the insightful questions. We have made the following clarifications and revisions:
> - We have corrected “an feasible” to “a feasible”.
> - We have corrected the notation from "$c(\dot)$" to the correct form "$c(\cdot)$".
> - In our formulation, $x\_{\\tau}^{*}$ denotes the policy at time $\\tau$ for the test trajectory $Y$, not for any calibration trajectory $Y^{(i)}$. It is a fixed quantity when computing $S\_{\\tau},S\_{\\tau}^{(1)},...,S\_{\\tau}^{(K+L)}$, and thus does not vary with $(i)$. Therefore, the notation $x\_{\\tau}^{\*}$ without a superscript $(i)$ is correct in this context.
>
> >**Limitations.**
>
> Thank you very much for your valuable comment. We would like to reiterate that the i.i.d. assumption on residual errors $\omega_{t}$ is stronger than necessary and can in fact be relaxed in our framework.
>
> Specifically, we do not require the prediction errors $\\omega\_{1},…,\\omega\_{T−1}$ within the same trajectory to be i.i.d.. Instead, we only require that across trajectories, the prediction error at a given time $t$ is i.i.d. between the test and calibration set. This means the residuals can exhibit arbitrary temporal dependence within a single trajectory, making our method broadly applicable to general time-series forecasting settings, including those with strong temporal correlation.
>
> We have clarified this point in the revised manuscript to avoid confusion. Once again, we sincerely thank the reviewer for highlighting this issue—it helped us further improve the rigor and generality of our assumptions. Should the reviewer have further questions, we would be happy to provide more detail or discussion.
>
> [1] *Human trajectory forecasting in crowds: A deep learning perspective. IEEE T-ITS, 2021.*
>
> [2] *Learning social etiquette: Human trajectory understanding in crowded scenes. ECCV, 2016.*

---

> ### Author Response · Authors · 2025-08-04
> **2nd Round Rebuttal by Authors**
>
> Thank you very much for your follow-up response. Below, we provide clarifications to all your concerns in two parts.
>
> **1. On the i.i.d. assumption.**
>
> We would like to further clarify that **relaxing the i.i.d. assumption does not require any substantial rework of the paper, nor does it affect the validity of the framework or the correctness of the theoretical results.** Furthermore, no modification to the existing proofs is needed.
>
> More specifically, the i.i.d. assumption on prediction errors $\\omega$ is used only once, in the proof of Lemma 4.2 (Appendix A.2). As stated at the beginning of the proof (line 470 in the original manuscript):
>
> > “We know that the random variables $\\omega\_{\\tau}, \\omega\_{\tau}^{(1)},...,\\omega\_{\tau}^{(K+L)}$ are i.i.d.”
>
> **This confirms that the proof only requires i.i.d. errors across trajectories at the same time step, and not i.i.d. errors within a single trajectory.** Thus, the relaxed assumption we described in our rebuttal—requiring only that $\\omega\_{\\tau}$ across trajectories are i.i.d. for each fixed $\\tau$—is sufficient and consistent with the original proof structure. We respectfully invite the reviewer to verify this directly in the original manuscript.
>
> **2. Overall risk guarantee for entire trajectory.**
>
> While this guarantee was not explicitly stated as a formal theorem in the original manuscript, it follows directly from existing lemmas and arguments already presented in the paper. To address your concern, we now provide the formal justification here for clarity, and will add it to the final version of the paper.
>
> **Theorem 5.4**. *(Overall risk guarantee for entire trajectory)* Given an overall risk tolerance $\\alpha$, if the posterior risk $\\beta\_{1:t}$ is calculated through Lemma 4.2, the risk $\\alpha\_{t+1:T}$ is allocated through ARA (14) or IRA (15), the planned state $x\_{t+1:T}^{\*}$ is a feasible solution of the TO problem (13) with $\\alpha\_{t+1:T}$, then the entire trajectory at time $t$ satisfies the overall risk guarantee $\\mathbb{P}\\{\\bigcap\_{\\tau=1}^{T}\\{c(x\_{\\tau}^{\*},Y\_{\\tau}) \\geq 0\\}\\} \\geq 1-\\alpha$.
>
> *Proof*: At each time $t$, the realized state $x\_{1:t}^{\*}$ is available. According to Lemma 4.2, we know that
>
> $$\\mathbb{P}\\{c(x\_{\\tau}^{\*}, Y\_{\\tau}) < 0\\} \\leq \\beta\_{\\tau},  \\forall \\tau = 1 ,…, t  \\tag{R1}$$
>
> For the planned state $x\_{t+1:T}^{\*}$, the corresponding risk allocations $\\alpha\_{t+1:T}$ are obtained either via ARA (14) or IRA (15). In both cases, they satisfy
>
> $$\\sum\_{\\tau =t+1}^{T} \\alpha\_{\\tau} \\leq \\alpha - \\sum\_{\\tau=1}^{t} \\beta\_{\\tau} \\tag{R2}$$
>
> We note that if the planned state $x\_{t+1:T}^{\*}$ is a feasible solution of the TO problem (13) with $\\alpha\_{t+1:T}$, the following inequality can be obtained through Lemma 4.1.
>
> $$\\mathbb{P}\\{c(x\_{\\tau}^{\*}, Y\_{\\tau}) \\geq 0\\} \\geq 1-\\alpha_{\\tau},  \\forall \\tau = t+1 ,…, T \\tag{R3}$$
>
> Finally, by combining the above three inequalities (R1), (R2), and (R3), together with Boole's inequality, we obtain
>
> $$\\mathbb{P}\\{\\bigcap\_{\\tau=1}^{T}\\{c(x_{\tau}^{\*},Y_{\tau}) \\geq 0\\}\\} \geq 1-\\alpha \\tag{R4}$$
>
> which means that the overall risk of the entire trajectory at time $t$, $x\_{1:T}^{\*}$, is below the risk tolerance $\\alpha$. Thus, Theorem 5.4 is proven.
>
> We sincerely hope that the above clarifications help dispel the reviewer’s misunderstanding and contribute to a more accurate evaluation of our manuscript. Should the reviewer have any further concerns, we would be happy to address them.

---

### Official Review · Reviewer_n484 · 2025-07-02

**Clarity:** 2
**Significance:** 2
**Originality:** 3
**Rating:** 4
**Confidence:** 4

**Summary:**

The paper discuss the application of conformal prediction (CP) for trajectory optimization. The authors identify a gap between the theoretical and application side of CP research and proposed a feedback-based CP frame work for srhinking horizon TO in uncertain environments with dynamic obstacles. The author provide theoretical analysis and numerical experiment in a simulation environment to show case the model is able to achieve good collision avoidance rate.

**Questions:**

- Why not using more powerful trajectory predictors for $g()$ in eqution 2?

**Ethical Concerns:**

["NO or VERY MINOR ethics concerns only"]

**Final Justification:**

Overall the paper has provide ample amount of theoratical groundings, the author provided more follow up experiments that address the reviewer's concern, hence raised the score.

**Limitations:**

The author mentioned limitation of the Fb-CP lies in the size of the calibaration data.
Consider the ample amount of trajectory prediciton works, the author should consider adopt a more sophisiticated algorithm to proper gate obstacle trajectory such as [1] and [2]. And consider apply this in a real world dataset such as [3].

[1].Salzmann, Tim, et al. "Trajectron++: Dynamically-feasible trajectory forecasting with heterogeneous data." Computer Vision–ECCV 2020: 16th European Conference, Glasgow, UK, August 23–28, 2020, Proceedings, Part XVIII 16. Springer International Publishing, 2020.
[2].Yuan, Ye, et al. "Agentformer: Agent-aware transformers for socio-temporal multi-agent forecasting." Proceedings of the IEEE/CVF international conference on computer vision. 2021.
[3].A. Robicquet, A. Sadeghian, A. Alahi, S. Savarese, Learning Social Etiquette: Human Trajectory Prediction In Crowded Scenes in European Conference on Computer Vision (ECCV), 2016.

**Paper Formatting Concerns:**

None Observed

**Quality:**

3

**Strengths And Weaknesses:**

### Strengths
- The paper is well motivated. The authors were able to provide ample amount intuition pointing out the gap in current reseach for comformal prediction for trajectory optimization problem
- The author provides comprehensive theoratical analysis on the effect of incorporating feeback information from realization of agent decision as time progresses.

### Weaknesses
- Despite the comprehensive theoratical analysis, the overall paper are quite dense and unfriendly to readers outside of the comformal prediction research domain. The author should consider reorganize the paper and put some of the lemma or corollary in the appendex.
- Consider the main contribution of the paper lies in providing a feedback-based CP for trajectory optimization, the author should consider put the algorithm in the appendix in the main paper. The author should also put more numerical results in the main paper to demonstrate the theoratical analysis.

---

> ### Author Rebuttal · Authors · 2025-07-29
>
> Thank you very much for your valuable comments, which have helped us to improve the quality of this manuscript.
>
> >**Weakness 1**
>
> We thank the reviewer for this valuable suggestion. To address the concern regarding accessibility and clarity, especially for readers who may not have a background in conformal prediction, we have made the following two major revisions.
>
> First, **we reorganized the structure of the main text and added informal, intuitive explanations after presenting mathematically dense definitions, theorems, or derivations**. This is intended to guide the reader through the logic of the proposed framework more smoothly, and to convey the key ideas in a more approachable way before introducing formal notation. Specifically:
> - After Lemma 4.1, we added an explanatory sentence directly to provide readers with an intuitive understanding of the lemma. The inserted text reads
> >“For a general collision avoidance chance constraint of the form $\\mathbb{P} \\{ \\Vert x_{\\tau} - Y_{\\tau} \\Vert - r \\geq 0 \\} \\geq 1 - \\alpha_{\\tau}$, Lemma 4.1 effectively converts it into a deterministic constraint that requires the distance between $x\_{\\tau}$ and the predicted location $\\hat{Y}\_{\\tau|t}$ to exceed the inflated radius derived from the prediction error, i.e., $\\Vert x\_{\\tau} - \\hat{Y}\_{\\tau|t} \\Vert - r - C\_{\\tau|t}^{1-\\alpha\_{\\tau}}  \\geq 0$."
> - After Lemma 4.2, we inserted a clarification sentence to convey the core idea behind the lemma in more intuitive terms. The newly added explanation states:
> >“Lemma 4.2 essentially computes the posterior probability by evaluating the fraction of calibration set $D\_{cal}^{2}$ samples whose actual trajectories would collide with the given realized position $x\_{\\tau}^{∗}$.”
> - After Lemma 5.1, we added an explanatory sentence to clarify the implication of the lemma's monotonicity result. Specifically, the inserted text reads:
> >“The monotonicity of $J^{∗}$ in Lemma 5.1 shows that increasing the allocated risk $\\alpha\_{\\tau}$ at any time step will strictly decrease the optimal cost of the trajectory optimizer. This insight enables our risk reallocation strategy: by reducing redundant risk at inactive time steps and reallocating it to active ones, we can lower the overall optimal cost.”
> - After Lemma 5.2, we added Remark 5.4 to provide an insightful explanation of the relationship between Lemma 5.2 and Lemma 4.2.
> >“**Remark 5.4.** One may notice that the calculation of $\\beta\_{\\tau}$ in Lemma 4.2 is similar to the computation of $\\widetilde{\\alpha}^{n}_{\\tau}$ in Lemma 5.2. This observation is correct. The key difference between the two lies in that the former utilizes the dataset $D\_{cal}^{2}$ independent with $D\_{cal}^{1}$ to achieve the coverage guarantee for $\\beta\_{\\tau}$. By contrast, as a step in solving (11), the latter does not need to consider the coverage guarantee and thus directly uses $D\_{cal}^{1}$. The use of different datasets results in the former providing probabilistic guarantee, while the latter achieves deterministic guarantee $(\\underline{\\alpha}\_{\\tau}^{n} \\leq \\alpha\_{\\tau}^{n})$.”
>
> We hope that these additions help readers outside the conformal prediction community understand both the motivation and the mechanism of our method more easily.
>
> Second, to further improve readability, **we moved some non-core theoretical results to the appendix**:
> - Lemma 3.2, which is a direct application of a lemma from [7], has been relocated to the appendix. While useful for completeness, it is not essential for understanding or implementing our proposed method.
> - Corollary 4.3, which provides a theoretical performance guarantee but is not directly involved in the operation of the framework, has also been moved to the appendix.
>
> We believe that these changes collectively improve the clarity, accessibility, and structure of the paper, and we hope they address the reviewer’s concerns.
>
> >**Weakness 2**
>
> We thank the reviewer for this insightful suggestion. We fully agree that the feedback-based CP framework is the core contribution of our work, and that the main paper would benefit from a more complete presentation of both the algorithm and supporting numerical results.
>
> Due to the strict 9-page limit of the NeurIPS main paper submission format, in the original manuscript, we placed the pseudocode and explanation of the proposed algorithm in Appendix B, and included only abbreviated experimental results in the main text, with full results provided in Appendix C. Fortunately, the final camera-ready version permitted by NeurIPS allows extending the main paper to 10 pages. As a result, following the reviewer’s suggestion, we revise the manuscript by
> - Move the pseudocode and its explanation from Appendix B into the main paper;
> - Move Table 3 from Appendix C.2 into the main text to provide more detailed numerical results and better support the theoretical analysis.
>
> We hope these updates will enhance the completeness and clarity of the main paper and better highlight the theoretical and practical contributions of our framework.
>
> >**Questions**
>
> We thank the reviewer for the helpful comment. We would like to emphasize that the proposed Fb-CP framework is general and modular, and is not restricted to the Social LSTM [6] used in our original experiments. **The trajectory predictor $g(\\cdot)$ can be replaced with any more powerful or complex predictor.** Indeed, using stronger predictors such as [1], [2], [4], or [5] should lead to improved prediction accuracy and therefore better downstream performance in trajectory optimization.
>
> **In the revised manuscript, we made this flexibility explicit by updating discussion and citing [1], [2], [4], and [5] around learning of $g(\\cdot)$ to include**
> > “*Naturally, readers may choose to adopt more powerful predictors for improved accuracy, such as Salzmann et al. [2020], Yuan et al. [2021], Salzmann et al. [2023], Yuan and Kitani [2020].*”
>
> Finally, to further validate this point, we conducted additional experiments using different predictors within our framework. These results are presented in a later section of our response and demonstrate that Fb-CP maintains superior performance regardless of the specific predictor used, and benefits even more from stronger predictors.
>
> >**Limitations**
>
> We sincerely thank the reviewer for this helpful suggestion. In response, we have conducted two additional sets of experiments to further validate the generality and real-world applicability of the proposed Fb-CP framework: (1) We evaluated our framework using more powerful trajectory predictors; (2) We applied our method to a real-world dataset, the Stanford Drone Dataset [3]. These experiments are detailed below.
>
> **(1) Using more advanced trajectory predictors.**
>
> We evaluated the performance of both our method (Fb-CP with ARA) and the baseline (S-CP) under three different prediction models:
> - The original Social LSTM predictor used in the original manuscript;
> - Two more sophisiticated algorithm, Trajectron++ [1] and AgentFormer[2].
>
> As shown in Table R1 below, the results demonstrate that more powerful trajectory predictors can enhance the performance of all methods by providing more accurate obstacle forecasts, and that Fb-CP consistently maintains superior performance over the baseline under both simple and advanced predictors.
>
> *Table R1: Average Cost of Different Methods using Different Trajectory Predictors*
> |  | S-CP | Fb-CP-ARA | Fb-CP-IRA |
> | :- | :-: | :-: | :-: |
> | Social LSTM (Original) | 17.30 | 15.49 | 2.973 |
> | Trajectron++ (Added) | 13.58 | 10.58 | 1.809 |
> | AgentFormer (Added) | 12.83 | 9.712 | 1.722 |
>
> **(2) Applying to a real-world dataset.**
>
> We further tested our method on the Stanford Drone Dataset. As shown in Table R2 below, Fb-CP-IRA outperforms all baselines and achieves the lowest average cost. In particular, it reduces the cost by at least 46% compared to S-CP. These results support the applicability of our method to practical, real-world environments with complex agent interactions.
>
> *Table R2: Average Cost of Different Methods on the Stanford Drone Dataset*
> | $\alpha$ / $\eta$ | CC | ACI-MP | S-CP | Fb-CP-ARA | Fb-CP-IRA |
> | :- | :-: | :-: | :-: | :-: | :-: |
> | 0.05/1000 | 95.51 | 40.02 | 38.58 | 30.58 | 20.81 |
> | 0.1/500 | 77.68 | 35.94 | 34.75 | 27.52 | 18.52 |
> | 0.15/100 | 41.27 | 32.57 | 30.48 | 23.75 | 15.58 |
> | 0.2/50 | 38.73 | 29.89 | 28.59 | 21.25 | 14.02 |
>
> We hope that these additional experiments effectively address the reviewer’s suggestions and demonstrate the flexibility and robustness of our proposed framework. If the reviewer has any further questions or concerns, we would be happy to clarify them.
>
> [1] *Tim Salzmann, Boris Ivanovic, Punarjay Chakravarty, and Marco Pavone. Trajectron++: Dynamically-feasible trajectory forecasting with heterogeneous data. European Conference on Computer Vision (ECCV), 2020.*
>
> [2] *Ye Yuan, Xinshuo Weng, Yanglan Ou, and Kris M Kitani. Agentformer: Agent-aware transformers for socio-temporal multi-agent forecasting. Proceedings of the IEEE/CVF international conference on computer vision. 2021.*
>
> [3] *Alexandre Robicquet, et al. Learning social etiquette: Human trajectory understanding in crowded scenes. European Conference on Computer Vision (ECCV), 2016.*
>
> [4] *Tim Salzmann, Hao-Tien Lewis Chiang, Markus Ryll, Dorsa Sadigh, Carolina Parada, and Alex Bewley. Robots that can see: Leveraging human pose for trajectory prediction. IEEE Robotics and Automation Letters, 2023.*
>
> [5] *Ye Yuan and Kris M. Kitani. Diverse trajectory forecasting with determinantal point processes. International Conference on Learning Representations, 2020.*
>
> [6] *Alexandre Alahi, et al. Social lstm: Human trajectory prediction in crowded spaces. IEEE Conference on Computer Vision and Pattern Recognition, 2016.*
>
> [7] *Ryan J Tibshirani, et al. Conformal prediction under covariate shift. Advances in neural information processing systems, 2019.*

---

> > ### Comment · Reviewer_n484 · 2025-08-05
> >
> > The reviewer believe that the author has addressed the major questions the reviewer raised and will raise the score accordingly.

---

> > > ### Author Response · Authors · 2025-08-05
> > > **Thank you very much for raising the score**
> > >
> > > Many thanks for raising the score and your kind support of our research work. We are very happy that our response has addressed your questions. Again, we greatly appreciate your valuable comments, which have significantly improved the quality of our paper.

---

### Official Review · Reviewer_PHGF · 2025-07-02

**Clarity:** 3
**Significance:** 2
**Originality:** 3
**Rating:** 4
**Confidence:** 5

**Summary:**

The paper proposes a Feedback-Based Conformal Prediction (Fb-CP) framework to improve trajectory optimization (TO) in uncertain environments. The main technical contribution is to create a closed-loop system where information from the realized past time step's trajectory is used to dynamically adjust the uncertainty prediction regions for future time steps.

**Questions:**

Please check the weaknesses for detail and address the points there.

**Ethical Concerns:**

["NO or VERY MINOR ethics concerns only"]

**Final Justification:**

This paper proposes a Feedback-Based Conformal Prediction (Fb-CP) framework to improve trajectory optimization (TO) in uncertain environments. The rebuttal has answered my questions, so I keep my relatively positive rating.

**Limitations:**

Yes

**Paper Formatting Concerns:**

There is no paper formatting issue.

**Quality:**

2

**Strengths And Weaknesses:**

Strengths:
1. The main technical novelty of the paper is the proposed feedback loop from the decision-making process back to the uncertainty quantification model (the CP prediction regions). It addresses a clear gap in existing literature where CP is used as a static, one-way "black box" in decision pipelines. This closed-loop approach is intuitive and makes practical sense for shrinking-horizon problems.

2. The theoretical coverage guarantee highlights the technical contribution. The paper proves that using the posterior risk calculation to adjust future prediction regions does not violate the fundamental coverage guarantees of Conformal Prediction (Lemma 4.2). This is crucial for safety-critical systemss such as autonomous driving.

Weaknesses:
1.However, the computational cost of the proposed Iterative Risk Allocation (IRA) algorithm may affect the application of the framework. The most performant version of the proposed method, Fb-CP-IRA, involves an iterative optimization process at each time step to allocate risk. As shown in the results Table 1 and Table 8, this process significantly increases computation time compared to the non-iterative methods. While the authors show that much of this cost is at the initial step and the subsequent steps are faster, it could still be a bottleneck for applications that require fast decision-making cycles (very high-frequency control).
2. The overall framework is very complex to implement, since it needs to combine a shrinking-horizon trajectory optimizer with a two-part conformal prediction system and an iterative risk allocation scheme. Implementing it correctly from scratch would be a non-trivial engineering effort compared to simpler, sequential CP methods in the literature. It needs to be justified with sufficient performance improvement for the system to apply this complex algorithm.
3. The proposed method and its theoretical guarantees relies on certain assumptions, such as the availability of a reasonably accurate dynamics model for obstacles (even if it has i.i.d. errors) and the ability to define a Lipschitz continuous constraint function. While common in the field, the performance of the method would still be sensitive to the quality of the underlying trajectory predictor.
4.The primary limitation, which the authors acknowledge, is the need to split the calibration dataset into two subsets (Dcal1 for forward prediction region generation and Dcal2 for backward posterior probability calculation). This means the method requires a larger calibration set than standard CP to achieve the same statistical power in each step. While the authors argue that large datasets are increasingly available, this could be a practical constraint in data-scarce domains.

---

> ### Author Rebuttal · Authors · 2025-07-29
>
> Thank you very much for your valuable comments, which have helped us to improve the quality of this manuscript.
>
> >**However, the computational cost of the proposed Iterative Risk Allocation (IRA) algorithm may affect the application of the framework. The most performant version of the proposed method, Fb-CP-IRA, involves an iterative optimization process at each time step to allocate risk. As shown in the results Table 1 and Table 8, this process significantly increases computation time compared to the non-iterative methods. While the authors show that much of this cost is at the initial step and the subsequent steps are faster, it could still be a bottleneck for applications that require fast decision-making cycles (very high-frequency control).**
>
> We thank the reviewer for raising this important point regarding computation time. We would like to emphasize that our contribution is a general Fb-CP framework for trajectory optimization, which supports **multiple risk allocation strategies**, including both **IRA** and **ARA**.
>
> While Fb-CP-IRA indeed achieves significant performance improvement by iteratively optimizing risk allocations, we fully agree that its computational demand may not be suitable for very high-frequency control. **For such settings, our framework also supports the non-iterative Fb-CP-ARA, which is extremely lightweight.** As shown in Table 2 of the original manuscript, Fb-CP-ARA incurs virtually no additional computation time compared to S-CP, yet still achieves at least a 7.21% reduction in average cost.
>
> Furthermore, to balance performance and computation efficiency, we propose a hybrid scheme (Appendix F) in which IRA is used only at the initial time step (offline), and ARA is used for all online steps. This approach eliminates online overhead of IRA, yet achieves a 66.56% cost reduction over pure ARA. The user can therefore flexibly select between IRA, ARA, or the hybrid variant, depending on their computational budget and real-time requirements.
>
> In summary, our framework allows users to flexibly choose between IRA, ARA, or a hybrid approach based on the computational constraints and real-time requirements of their specific application. In particular, ARA provides a fast and practical solution, incurring almost no extra runtime compared to baselines, while still benefiting from feedback-based prediction adjustment.
>
> >**The overall framework is very complex to implement, since it needs to combine a shrinking-horizon trajectory optimizer with a two-part conformal prediction system and an iterative risk allocation scheme. Implementing it correctly from scratch would be a non-trivial engineering effort compared to simpler, sequential CP methods in the literature. It needs to be justified with sufficient performance improvement for the system to apply this complex algorithm.**
>
> We thank the reviewer for highlighting this important concern. We agree that integrating feedback-based conformal prediction with trajectory optimization introduces additional system complexity compared to sequential CP methods. However, we believe the significant and consistent performance improvements justify this complexity.
>
> As shown in Appendix C of the original manuscript, **we conducted extensive experiments across three representative dynamical systems**—a kinematic vehicle [1], a 3D linear quadrotor [2], and a dynamic bicycle model [3]—and **compared against four state-of-the-art baselines**: CC [1], ACI-MP [2], RF-CP [4], and S-CP [5]. The proposed Fb-CP-ARA consistently achieved substantial cost reductions over the S-CP baseline, including:
> - 78.5% reduction on the kinematic vehicle model,
> - 58.5% reduction on the 3D quadrotor model,
> - 40.2% reduction on the dynamic bicycle model.
>
> To show performance improvement in real-world problems, **we evaluated our method on the Stanford Drone Dataset [6]**. As shown in Table R1 below, Fb-CP-IRA outperforms all baselines and achieves the lowest average cost. In particular, it reduces the cost by at least 46% compared to S-CP. Note that RF-CP is not included as a baseline due to its reliance on mixed-integer variables, which combined with nonlinear dynamics, results in intractable trajectory optimization.
>
> *Table R1: Average Cost of Different Methods on the Stanford Drone Dataset*
> | $\alpha$ / $\eta$ | CC | ACI-MP | S-CP | Fb-CP-ARA | Fb-CP-IRA |
> | :- | :-: | :-: | :-: | :-: | :-: |
> | 0.05/1000 | 95.51 | 40.02 | 38.58 | 30.58 | 20.81 |
> | 0.1/500 | 77.68 | 35.94 | 34.75 | 27.52 | 18.52 |
> | 0.15/100 | 41.27 | 32.57 | 30.48 | 23.75 | 15.58 |
> | 0.2/50 | 38.73 | 29.89 | 28.59 | 21.25 | 14.02 |
>
> Finally, to reduce the engineering effort and support adoption, we will release the full implementation of our method upon paper acceptance. We hope this will make our framework easier to use and build upon for both researchers and practitioners.
>
> >**The proposed method and its theoretical guarantees relies on certain assumptions, such as the availability of a reasonably accurate dynamics model for obstacles (even if it has i.i.d. errors) and the ability to define a Lipschitz continuous constraint function. While common in the field, the performance of the method would still be sensitive to the quality of the underlying trajectory predictor.**
>
> We thank the reviewer for the insightful comment. We would like to clarify that the core novelty of our work lies in the Feedback-based Conformal Prediction (Fb-CP) framework, rather than in the design of a specific trajectory predictor. **Our method can work with any trajectory prediction module, including classical models or modern deep learning-based predictors.**
>
> We fully agree that the quality of the trajectory predictor can affect the performance of the resulting trajectory optimization, as it influences the prediction error and thus the conservativeness of the conformal prediction region. **This sensitivity, however, is inherent to all CP-based decision-making methods, not unique to our framework.** As shown in Table 3 of the original manuscript, when using the same predictor, our method consistently outperforms existing baselines across, which highlights the strength of our feedback mechanism regardless of the predictor accuracy.
>
> To further validate that our performance gains are independent of the choice of predictor, we conducted additional experiments comparing our method and S-CP under different predictors. As shown in Table R2 below, when switching to more advanced predictors Trajectron++ [7] and AgentFormer [8], both our method and the baseline see improved performance, but our framework continues to outperform the baseline under all tested predictors.
>
> *Table R2: Average Cost of Different Methods using Different Trajectory Predictors*
> |  | S-CP | Fb-CP-ARA | Fb-CP-IRA |
> | :- | :-: | :-: | :-: |
> | Social LSTM [9] | 17.30 | 15.49 | 2.973 |
> | Trajectron++ [7] | 13.58 | 10.58 | 1.809 |
> | AgentFormer [8] | 12.83 | 9.712 | 1.722 |
>
> This confirms that while the choice of predictor affects the absolute performance—as expected—our Fb-CP framework consistently maintains higher performance compared to the baseline method regardless of predictor quality. Additionally, in practice, **our framework can be directly integrated with more sophisticated predictors** as needed in real-world applications, enabling further performance gains without changing the core structure of our algorithm.
>
> >**The primary limitation, which the authors acknowledge, is the need to split the calibration dataset into two subsets (Dcal1 for forward prediction region generation and Dcal2 for backward posterior probability calculation). This means the method requires a larger calibration set than standard CP to achieve the same statistical power in each step. While the authors argue that large datasets are increasingly available, this could be a practical constraint in data-scarce domains.**
>
> Thank you very much for your valuable comment. Frist, we would like to clarify that in all our experiments, the total amount of calibration data used by Fb-CP and the baselines is exactly the same. Although the split reduces the number of samples used in each stage (forward prediction or backward feedback), the performance gains brought by feedback-based adjustment significantly outweigh the minor statistical loss from halving the calibration size.
>
> To directly address the reviewer’s concern, we conducted additional experiments under a data-scarce regime with only $N=400$ calibration samples. As shown in Table R3 below, although all methods see a decrease in performance under limited data, Fb-CP still achieves substantially lower cost than S-CP and remains safety. This highlights the robustness and practical effectiveness of our method, even when the calibration dataset is small.
>
> *Table R3: Average Cost of Different Methods with the Calibration Data Size of $N=400$*
> |  | S-CP | Fb-CP-ARA | Fb-CP-IRA |
> | :- | :-: | :-: | :-: |
> | Average cost | 18.03 | 16.82 | 3.521 |
> | Average computation time | 0.078 | 0.081 | 0.122 |
> | Collision avoidance rate | 88.9% | 89.5% | 91.7% |
>
> [1] *Conformal decision theory: Safe autonomous decisions from imperfect predictions. ICRA, 2024.*
>
> [2] *Adaptive conformal prediction for motion planning among dynamic agents. L4DC, 2023.*
>
> [3] *Wasserstein distributionally robust motion control for collision avoidance using conditional value-at-risk. IEEE T-RO, 2021.*
>
> [4] *Recursively feasible shrinking-horizon mpc in dynamic environments with conformal prediction guarantees. L4DC, 2024.*
>
> [5] *Safe planning in dynamic environments using conformal prediction. IEEE R-AL, 2023.*
>
> [6] *Learning social etiquette: Human trajectory understanding in crowded scenes. ECCV, 2016.*
>
> [7] *Trajectron++: Dynamically-feasible trajectory forecasting with heterogeneous data. ECCV, 2020.*
>
> [8] *Agentformer: Agent-aware transformers for socio-temporal multi-agent forecasting. ICCV. 2021.*
>
> [9] *Social lstm: Human trajectory prediction in crowded spaces. CVPR, 2016.*

---

> > ### Comment · Reviewer_PHGF · 2025-08-04
> >
> > Thanks for the authors' response, it has addressed my concerns. I will keep my original score that is inclined to acceptance of this work.

---

> > > ### Author Response · Authors · 2025-08-04
> > > **Thank you very much**
> > >
> > > Many thanks for your recognition and support of our research work. We are very happy that our response has addressed your concerns. Again, we greatly appreciate your valuable comments, which have significantly improved the quality of our paper.

---

### Official Review · Reviewer_62Tj · 2025-07-04

**Clarity:** 3
**Significance:** 3
**Originality:** 3
**Rating:** 5
**Confidence:** 3

**Summary:**

This paper proposes a feedback-based conformal prediction framework for trajectory optimization with safety constraints that dynamically adjusts prediction regions using realised trajectories, improving decision performance while preserving coverage guarantees. It proves that the proposed algorithm (Fb-CP) offers guarantees for performance improvement and provide a convergence analysis of the algorithm.

**Questions:**

## Questions

l.116 - dynamica - dynamics
l.133 - I assume that it's actually not according to eq.2 based on how you explain that g() is modelled by an LLM. Do you mean the ground truth dynamics here in assump. 2? You should clarify this.

eq3, please fix the formatting here.

l.152 Shouldn't the superscript start from 0?

l.168 -  Explain the abbreviation Fb-CP.

eq. 15a - 15d, fix alignment


## General questions

For the real-world, we cannot expect that we have full-observability of obstacles. An example of this is a car in traffic, we see the cars moving but they follow very complex dynamics with a human involved that are difficult to predict. How does your algorithm deal with those situations / how does this influence the uncertainty quantification via CP?

**Ethical Concerns:**

["NO or VERY MINOR ethics concerns only"]

**Final Justification:**

I think that this is technically solid work introducing an uncertainty-aware algorithm for trajectory optimization that utilizes conformal prediction. For this reason, I would argue for the acceptance of the paper.

**Limitations:**

yes

**Quality:**

4

**Strengths And Weaknesses:**

## Strengths

Safe trajectory optimization algorithms are an active area of research, much work has been seen in combining machine learning methods and OC methods to arrive to better algorithms, especially when considering uncertainty quantification where research has been done in using generative models such as flow, gan's etc. but as well to use zero-order sampling techniques for evaluating safety constraint violation. Conformal prediction offers a training-free alternative to quantifying uncertainty in the models prediction with strong guarantees.

While the introduction of conformal prediction has been explored in the trajectory optimization setting, the paper offers a thorough analysis with convergence guarantees that improves the usage of information based on the past encountered states. This will enable principled usage of conformal prediction in a feedback loop, which is very useful for application.

The paper also explores different risk allocation approaches,  average risk allocation and iterative risk allocation.

The paper provides bounds on the posterior violation probability as well as performance guarantees showing that the proposed method is at  least as good as the sequential method from Lindemann [2023], but in practice the proposed method performs even better experimentally.


## Weaknesses

The issue with conformal prediction is that one needs good coverage for good guarantees, which is a condition that is hard to satisfy in high-dimensional spaces.

The experimental setting for verifying the algorithm is very limited, as the environments are low-dimensional, the application to real-world problems is questionable.  Nevertheless, I believe that the proposed algorithm and theoretical insights are useful for the community.

---

> ### Author Rebuttal · Authors · 2025-07-29
>
> Thank you very much for your valuable comments, which have helped us to improve the quality of this manuscript.
>
> >**The issue with conformal prediction is that one needs good coverage for good guarantees, which is a condition that is hard to satisfy in high-dimensional spaces.**
> >
> >**The experimental setting for verifying the algorithm is very limited, as the environments are low-dimensional, the application to real-world problems is questionable. Nevertheless, I believe that the proposed algorithm and theoretical insights are useful for the community.**
>
> Thank you very much for your kind recognition of the proposed method. First, we would like to clarify that our method is specifically designed for Trajectory Optimization (TO) tasks, where the problem dimension is inherently low (typically 2D or 3D) due to the physical configuration of robots and vehicles. In such settings, Conformal Prediction (CP) is practical and effective.
>
> Second, while our problems are in low-dimensional spaces (by the nature of TO), the experimental evaluation is actually not limited. In the original paper (Appendix C), we conducted extensive experiments on three distinct dynamical systems—a kinematic vehicle [1], a 3D linear quadrotor [2], and a dynamic bicycle model [3]—and compared against four state-of-the-art baselines: Conformal Control (CC) [1], ACI for Motion Planning (ACI-MP) [2], Recursively Feasible CP (RF-CP) [4], and Sequential CP (S-CP) [5]. In all cases, our proposed Fb-CP framework consistently demonstrates superior performance.
>
> Moreover, to further showcase the potential of our method in real-world problems, we have additionally evaluated it on the real-world Stanford Drone Dataset [6]. As shown in Table R1 below, Fb-CP with IRA outperforms all baselines and achieves the lowest average cost. In particular, it reduces the cost by at least 46% compared to S-CP. Note that RF-CP is not included as a baseline because its formulation introduces mixed-integer variables through nonconformity score definitions, which—when combined with the nonlinear vehicle dynamics—leads to highly complex trajectory optimization problems with prohibitively long computation times. These results support the applicability of our method to real-world problems.
>
> *Table R1: Average Cost of Different Methods on the Real-World Stanford Drone Dataset*
> | $\alpha$ / $\eta$ | CC | ACI-MP | S-CP | Fb-CP-ARA | Fb-CP-IRA |
> | :---- | :----: | :----: | :----: | :----: | :----: |
> | 0.05/1000 | 95.51 | 40.02 | 38.58 | 30.58 | 20.81 |
> | 0.1/500 | 77.68 | 35.94 | 34.75 | 27.52 | 18.52 |
> | 0.15/100 | 41.27 | 32.57 | 30.48 | 23.75 | 15.58 |
> | 0.2/50 | 38.73 | 29.89 | 28.59 | 21.25 | 14.02 |
>
> Furthermore, to directly evaluate the effectiveness of our framework in higher-dimensional settings, we conducted an additional numerical case study involving a generic dynamical system under a shrinking-horizon setup with higher-dimensional state spaces. As shown in Table R2 below, our feedback-based CP framework consistently outperforms baseline methods while maintaining safety, even in these higher-dimensional scenarios. These results validate the applicability and effectiveness of our approach in high-dimensional problems.
>
> *Table R2: Average Cost and Collision Avoidance Rate of Different Methods ($\\alpha=0.2$) under Varying Dimensions*
> |  | Dimension | S-CP | Fb-CP-ARA | Fb-CP-IRA |
> | :- | :-: | :-: | :-: | :-: |
> | **Average cost** | 5 | 44.52 | 39.67 | 30.25 |
> |  | 10| 72.66 | 65.33 | 60.18 |
> | **Collision avoidance rate** | 5 | 92.5% | 92.1% | 93.5% |
> |  | 10 | 91.2% | 93.5% | 91.4% |
>
> >**l.116 - dynamica - dynamics l.133 - I assume that it's actually not according to eq.2 based on how you explain that g() is modelled by an LLM. Do you mean the ground truth dynamics here in assump. 2? You should clarify this.**
>
> We sincerely thank the reviewer for pointing out the typo on line 116 ("dynamica")—we have corrected it to "dynamics" in the revised manuscript.
>
> Regarding the comment on line 133, the reviewer is absolutely correct. In Assumption 3.1, the calibration dataset $D_{cal}$ is generated using the true underlying obstacle dynamics, not the learned model $g(\cdot)$. As clarified, $g(\cdot)$ is trained from a separate dataset and only used at test time to generate predictions $\hat{Y}_{\tau|t}$. We have updated the statement in Assumption 3.1 accordingly to avoid confusion.
>
> >**l.eq3, please fix the formatting here.**
>
> Thank you for your comment. We have revised the formatting of eq. (3) to improve clarity and readability. In particular, the constraints and their applicable time indices are now properly aligned and consistently presented.
>
> >**l.152 Shouldn't the superscript start from 0?**
>
> Thank you for your comment. In our manuscript, the superscript $(i)$ is used to index the calibration samples, while the symbol without a superscript (i.e., $R$) refers to the test data point. Using superscript $(0)$ for the test point is certainly a valid and alternative convention. Following the reviewer’s suggestion, we have updated the notation in the revised manuscript to use $R^{(0)}$ for the test point, and $R^{(1)},…,R^{(N)}$ for the calibration set, for improved clarity and consistency.
>
> >**l.168 - Explain the abbreviation Fb-CP.**
>
> Thank you for pointing this out. We have updated the manuscript to clarify the meaning of “Fb-CP” as “**F**eed**b**ack-based **C**onformal **P**rediction” at line 168.
>
> >**eq. 15a - 15d, fix alignment**
>
> Thank you for your comment. Similar to our revision of eq. (3), we have fixed the formatting of eq. (15) in the revised manuscript to improve alignment.
>
> >**For the real-world, we cannot expect that we have full-observability of obstacles. An example of this is a car in traffic, we see the cars moving but they follow very complex dynamics with a human involved that are difficult to predict. How does your algorithm deal with those situations / how does this influence the uncertainty quantification via CP?**
>
> We fully agree with the reviewer that full observability of obstacle dynamics is unrealistic in real-world scenarios, especially in human-in-the-loop systems such as traffic. Our framework is in fact designed to accommodate such partial observability.
>
> First, in our formulation, $Y_{t}$ does not need to represent the complete internal dynamics of obstacles. It can simply represent any observable states relevant to safety—such as position and velocity—which are commonly available from sensors. The learned prediction model $g(\cdot)$ can be interpreted more precisely as a function of past observations within a time window:
>
> $Y_{t} = g(Y_{t-h},...,Y_{t-1}) + \omega_{t}$
>
> where $h$ is the window length. This allows for the use of flexible, data-driven time-series forecasting models such as Social-LSTM [7], Trajectron++ [8] or AgentFormer [9]. Therefore, our method can naturally operate under partial observability of the obstacle state.
>
> Second, while partial observability does not invalidate the safety guarantees provided by conformal prediction, it does influence performance. Specifically, limited observability may increase the prediction error $\omega_{t}$, leading to wider conformal prediction regions. This increased conservatism in the confidence region can result in less efficient trajectories, as the planner needs to avoid these larger obstacle position confidence regions
>
> In summary, our framework remains applicable and safe under partial observability, though its performance may degrade gracefully depending on the quality of the available observations and the predictive model used. We have also validated the effectiveness of our approach on the real-world Stanford Drone Dataset [6], demonstrating its practical applicability under realistic observation conditions.
>
> [1] *Jordan Lekeufack, Anastasios N Angelopoulos, Andrea Bajcsy, Michael I Jordan, and Jitendra Malik. Conformal decision theory: Safe autonomous decisions from imperfect predictions. IEEE International Conference on Robotics and Automation (ICRA), 2024.*
>
> [2] *Anushri Dixit, Lars Lindemann, Skylar X Wei, Matthew Cleaveland, George J Pappas, and Joel W Burdick. Adaptive conformal prediction for motion planning among dynamic agents. Learning for Dynamics and Control Conference, 2023.*
>
> [3] *Astghik Hakobyan and Insoon Yang. Wasserstein distributionally robust motion control for collision avoidance using conditional value-at-risk. IEEE Transactions on Robotics, 2021.*
>
> [4] *Charis Stamouli, Lars Lindemann, and George Pappas. Recursively feasible shrinking-horizon mpc in dynamic environments with conformal prediction guarantees. Learning for Dynamics and Control Conference, 2024.*
>
> [5] *Lars Lindemann, Matthew Cleaveland, Gihyun Shim, and George J Pappas. Safe planning in dynamic environments using conformal prediction. IEEE Robotics and Automation Letters, 2023.*
>
> [6] *Alexandre Robicquet, Amir Sadeghian, Alexandre Alahi, and Silvio Savarese. Learning social etiquette: Human trajectory understanding in crowded scenes. European Conference on Computer Vision (ECCV), 2016.*
>
> [7] *Alexandre Alahi, Kratarth Goel, Vignesh Ramanathan, Alexandre Robicquet, Li Fei-Fei, and Silvio Savarese. Social lstm: Human trajectory prediction in crowded spaces. IEEE Conference on Computer Vision and Pattern Recognition, 2016.*
>
> [8] *Tim Salzmann, Boris Ivanovic, Punarjay Chakravarty, and Marco Pavone. Trajectron++: Dynamically-feasible trajectory forecasting with heterogeneous data. European Conference on Computer Vision (ECCV), 2020.*
>
> [9] *Ye Yuan, Xinshuo Weng, Yanglan Ou, and Kris M Kitani. Agentformer: Agent-aware transformers for socio-temporal multi-agent forecasting. Proceedings of the IEEE/CVF international conference on computer vision. 2021.*

---

> > ### Author Response · Authors · 2025-08-07
> > **Follow-up on Our Rebuttal Response to Reviewer**
> >
> > First of all, we greatly appreciate your recognition and support of our research work. As the deadline of Author-Reviewer discussion approaches, we would like to kindly follow up to see if our rebuttal has addressed your concerns. If there are any remaining questions or clarifications needed, we would be more than happy to provide further responses.
> >
> > Thank you very much for your time and consideration.

---

> > > ### Comment · Reviewer_62Tj · 2025-08-09
> > > **Thank you for your response**
> > >
> > > I thank yhe authors for addressing my comments, I don't have any further concerns. I think the work is interesting for this conference venue and I will argue for its acceptance.

---

> > > > ### Author Response · Authors · 2025-08-09
> > > > **Thank you so much for your kind support!**
> > > >
> > > > We sincerely appreciate your kind support and recognition of our research work. We are very happy that our response has addressed your concerns. Again, many thanks for your insightful and valuable comments, which have significantly improved the quality of our paper.

---

### Note · Authors · 2025-08-11

First of all, we would like to express our deepest gratitude to the Area Chairs and all reviewers for their time, effort, and constructive feedback throughout the review process. We are pleased to report that we have addressed all concerns raised by the reviewers, and each reviewer has acknowledged that their concerns have been resolved. The comments provided have greatly improved both the quality and clarity of our work.

In particular, following the reviewers’ suggestions, we have:

- Verified the effectiveness of our framework in higher-dimensional settings;
- Demonstrated its robustness and effectiveness in data-scarce domains;
- Validated the effectiveness and safety of the method on the real-world Stanford Drone Dataset;
- Evaluated the method with more advanced trajectory predictors, showing that it consistently achieves superior performance over baseline methods;
- Relaxed an unnecessarily strong i.i.d. assumption without requiring any substantial rework of the paper.

We sincerely thank the Area Chairs again for their guidance and the reviewers for their thoughtful concerns, which have substantially strengthened our work.

---

### Decision · Program_Chairs · 2025-09-17

**Decision:**

Accept (poster)

**Comment:**

(a) Summary

This paper proposes a feedback-based conformal prediction (Fb-CP) framework for trajectory optimization under safety constraints in uncertain, dynamic environments. The core idea is to dynamically adjust the conformal prediction regions for future trajectory steps by incorporating information from realized past trajectories. In doing so, the method could improve decision performance while maintaining statistical coverage guarantees. The authors provide theoretical analysis, including convergence guarantees and bounds on posterior violation probability, and compare their method to existing sequential conformal prediction approaches. Experimental results are limited to low-dimensional simulated environments, but they demonstrate improved empirical performance and safety (collision avoidance) over baselines.

(b) Strengths

1. The paper addresses a clear gap in the literature by introducing a closed-loop feedback mechanism into conformal prediction for trajectory optimization, moving beyond the standard static application of CP. The authors provide rigorous theoretical analysis, including coverage guarantees and convergence proofs, which are crucial for safety-critical applications.
2. The method is well-motivated for real-world problems where uncertainty quantification and safety are paramount, such as autonomous driving.
3.  The paper explores different risk allocation strategies (average and iterative) and provides bounds on violation probabilities, showing both theoretical and empirical improvements over prior work.

(c) Weaknesses

- The experiments are restricted to low-dimensional, simulated environments. The applicability to real-world, high-dimensional problems remains untested, and the scalability of the approach is unclear.
- The iterative risk allocation (IRA) variant, while performant, incurs significant computational overhead, which may limit its use in real-time or high-frequency control settings.
- The proposed framework is complex, requiring integration of trajectory optimization, conformal prediction, and risk allocation, which may hinder adoption without clear evidence of substantial performance gains.
- The method assumes access to accurate dynamics models and requires splitting the calibration dataset, increasing data demands. This could be problematic in data-scarce domains.
- The trajectory prediction component could leverage more sophisticated models from recent literature, and experiments on real-world datasets would strengthen the empirical case.

The paper receives positive scores from four reviewers: One reviewer recommends an accept, three reviewers recommend a borderline accept. The reviewers reach a consensus that the paper is a technically solid and original contribution to the intersection of conformal prediction and trajectory optimization. While the experimental validation is limited and the method is complex (complicated), the theoretical insights and the potential for impact in safety-critical applications outweigh these concerns.